# Task complexity interacts with state-space uncertainty in the arbitration between model-based and model-free learning

Dongjae Kim [1,2], Geon Yeong Park[1], John P. O'Doherty[3,4,7]* & Sang Wan Lee[1,2,5,6,7]*

It has previously been shown that the relative reliability of model-based and model-free reinforcement-learning (RL) systems plays a role in the allocation of behavioral control between them. However, the role of task complexity in the arbitration between these two strategies remains largely unknown. Here, using a combination of novel task design, computational modelling, and model-based fMRI analysis, we examined the role of task complexity alongside state-space uncertainty in the arbitration process. Participants tended to increase model-based RL control in response to increasing task complexity. However, they resorted to model-free RL when both uncertainty and task complexity were high, suggesting that these two variables interact during the arbitration process. Computational fMRI revealed that task complexity interacts with neural representations of the reliability of the two systems in the inferior prefrontal cortex.

[1] Department of Bio and Brain Engineering, Korea Advanced Institute of Science Technology (KAIST), Daejeon 34141, Republic of Korea. [2] Program of Brain and Cognitive Engineering, KAIST, Daejeon 34141, Republic of Korea. [3] Computation & Neural Systems, California Institute of Technology, Pasadena, CA 91125, USA. [4] Division of Humanities and Social Sciences, California Institute of Technology, Pasadena, CA 91125, USA. [5] KI for Health Science Technology, KAIST, Daejeon 34141, Republic of Korea. [6] KI for Artificial Intelligence, KAIST, Daejeon 34141, Republic of Korea. [7]These authors jointly supervised this work: John P. O'Doherty, Sang Wan Lee. *email: jdoherty@caltech.edu; swlee@kaist.ac.kr

It has been suggested that two distinct mechanisms exist for controlling instrumental actions: a model-free RL system that learns values for actions based on the history of rewards obtained on those actions[1–3] and a model-based RL (MB) system that computes action values flexibly based on its knowledge about state-action-state transitions incorporated into an internal model of the structure of the world[4,5]. These two systems have different relative advantages and disadvantages. While model-free RL can be computationally cheap and efficient, it achieves this at the cost of a lack of flexibility, thereby potentially exposing the agent to inaccurate behavior following a change in either goal values and/or environmental contingencies[1,2,6]. On the other hand, MB RL is computationally expensive as it requires active computation of expected values and planning, but retains flexibility in that it can rapidly adjust a behavioral policy following a change in goal values or environmental contingencies. The theoretical trade-offs between these two systems has, alongside empirical evidence for the parallel existence of these two modes of computation in the behavior of animals and humans[2,7–28], prompted interest in elucidating how the trade-off between these systems is actually managed in the brain.

One influential proposal is that there exists an arbitration process that allocates control to the two systems[6,16,29]. One specific version of this theory suggests that estimates about the uncertainty in the predictions of the two systems mediates the trade-off between the respective controllers, such that under situations where the model-free system has unreliable predictions, the model-based system is assigned greater weight over behavior, while under situations where the model-free system has more accurate predictions, it will be assigned greater behavioral control[6,30].

A challenge in building a computationally and biologically plausible theory of arbitration is that arbitration itself should not be computationally expensive so as to render relative savings in computational cost associated with being model-free vs between model-based moot. To this end, practical computational theories of arbitration have examined computationally cheap approximations that might be utilized to mediate the arbitration process. According to Lee et al.[30], uncertainty is approximated via a mechanism that tracks cumulative predictions errors induced in the two systems. Model-free uncertainty is approximated via the average accumulation of reward-prediction errors (RPEs), while model-based uncertainty is approximated via the accumulation of errors in state prediction (so-called state-prediction errors (SPEs)).

Utilizing this framework, Lee et al.[30] examined the neural correlates of arbitration. In that study, a region of bilateral ventrolateral prefrontal cortex was found to track the reliabilities (an approximation of the inverse of uncertainty) in the predictions of the two systems, consistent with a role for this brain region in the arbitration process itself. However, the relative uncertainty or reliability in the predictions of the two systems is only one-component of the trade-off between the two controllers. Another equally important element of this trade-off is the relative computational cost of engaging in model-based control.

The goal of the present study is to investigate the role of computational cost in the arbitration process, alongside relative uncertainty. We experimentally manipulated computational cost by means of adjusting the complexity of the planning problem faced by the model-based controller. This was achieved by subjecting participants to a multi-step Markov Decision Problem in which the number of actions available in each state was experimentally manipulated. In one condition, which we called low complexity, two actions were available, while in another condition, which we called high complexity, four actions were available. In addition to manipulating complexity, we also, as in our

previous study[30], manipulated the uncertainty in the state-space, by utilizing a state-transition structure in the Markov decision process (MDP) that invoked high levels of SPE (i.e. one where the transitions are maximally uncertainty), and a transition structure where the transitions are either high or low in uncertainty. Thus, we manipulated two variables in a factorial design: state-space complexity (low vs high), and state-space uncertainty (low vs high).

This design allowed us to investigate the ways in which uncertainty and complexity interact to drive the arbitration process, thereby allowing us to assess the interaction between (model-based) reliability and at least one simple proxy of computational cost. While participants were undergoing this novel behavioral task, we also simultaneously measured brain activity with functional magentic resonance imaging (fMRI). This allowed us to investigate the contribution of state-space complexity alongside state-space uncertainty in mediating arbitration at both behavioral and neural levels. In order to accommodate the effects of task complexity in the arbitration process, we extended our previous arbitration model to endow this arbitration scheme with the capability of adjusting the arbitration process as a function of complexity. On the neural level, we focused on the ventrolateral prefrontal cortex as our main brain region of interest, given this was the main region implicated in arbitration in our previous study. We hypothesized that an arbitration model sensitive to both the complexity of the state-space and the degree of uncertainty in state-space transitions would provide a better account of behavioral and fMRI data than an arbitration model sensitive only to state-space uncertainty. We further hypothesized that under high complexity, the model-based controller would be selected against, because such complexity would overwhelm the planning requirements of the model-based system, forcing participants to rely instead on model-free control. Our findings support our first hypothesis, and partially support our second hypothesis.

## Results

**Markov decision task with varying degree of complexity**. To investigate the role of uncertainty and complexity in arbitration control, we designed a novel two-stage MDP task (Fig. 1a), in which we systematically manipulated two task variables across blocks of trials, state-transition uncertainty, and state-space complexity (Fig. 1b). The amount of state-transition uncertainty is controlled by means of the state-action-state transition probability. The state-action-state transition probability varies between the two conditions: high uncertainty (0.5 vs 0.5) and low uncertainty (0.9 vs 0.1). Note that our high-uncertainty condition is intended to maximize variability in the amount of SPE, as opposed to making participants perfectly learn the state-transition probabilities (0.5, 0.5). In fact, it would be more challenging to test effects on behavior of more moderate uncertainty conditions, such as (0.7, 0.3) or (0.6, 0.4), within a relatively short blocks of trials.

Switching between the two uncertainty conditions is designed to induce change in the average amount of SPEs, thereby influencing the reliability of the MB system. For instance, the high uncertainty condition will elicit a large amount of SPEs, essentially resulting in a decrement in MB prediction performance. In the low uncertainty condition, the SPE will decrease or stay low on average as the MB refines an estimate of the state-action-state transition probabilities. On the other hand, the performance of MF is less affected by the amount of state-transition uncertainty[6]. The second variable, the number of available choices, is intended to manipulate task complexity. The total number of available choices is two and four in the low and

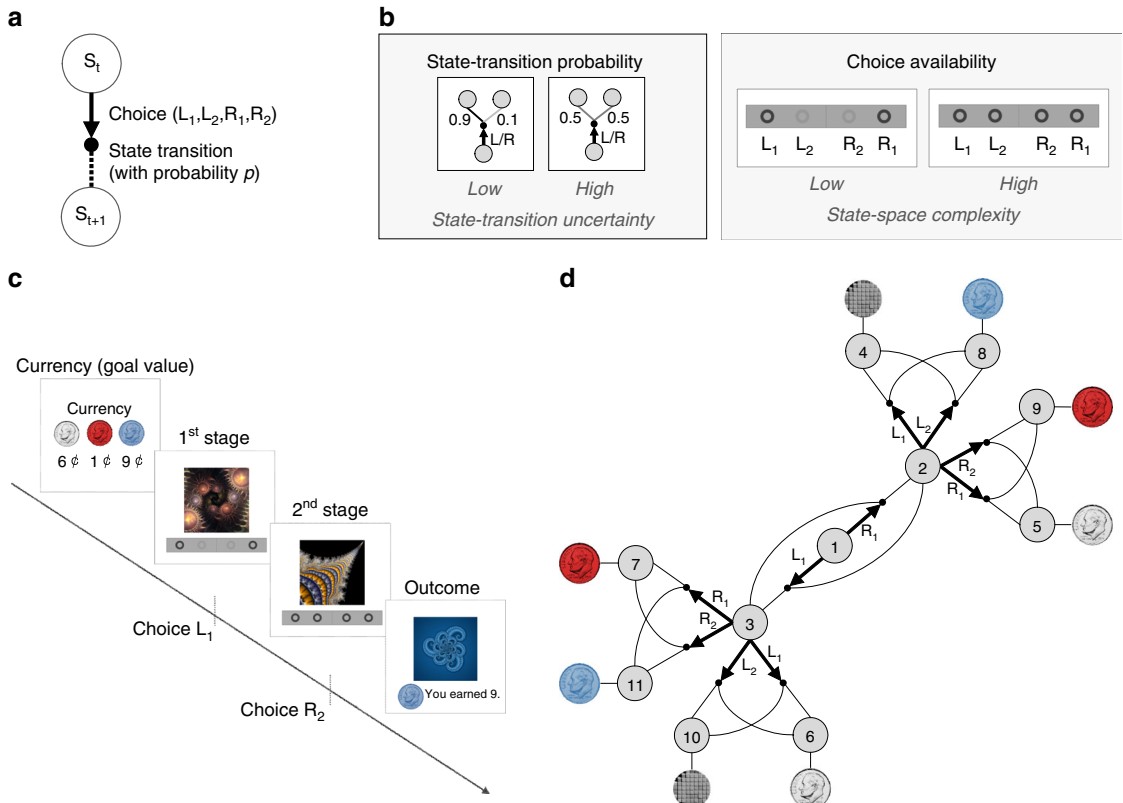

**Fig. 1 Task design. a** Two-stage Markov decision task. Participants choose between two to four options, followed by a transition according to a certain state-transition probability $p$, resulting in participants moving from one state to the other. The probability of successful transitions to a desired state is proportional to the estimation accuracy of the state-transition probability, and it is constrained by the entropy of the true probability distribution of the state-transition. For example, the probability of a successful transition to a desired state cannot exceed 0.5 if $p=(0.5, 0.5)$ (the highest entropy case). **b** Illustration of experimental conditions. (Left box) A low and high state-transition uncertainty condition corresponds to the state-transition probability $p = (0.9, 0.1)$ and $(0.5, 0.5)$, respectively. (Right box) The low and high state-space complexity condition corresponds to the case where two and four choices are available, respectively. In the first state, only two choices are always available, in the following state, two or four options are available depending on the complexity condition. **c** Participants make two sequential choices in order to obtain different colored tokens (silver, blue, and red) whose values change over trials. On each trial, participants are informed of the "currency", i.e. the current values of each token. In each of the subsequent two states (represented by fractal images), they make a choice by pressing one of available buttons (L1, L2, R1, R2). Choice availability information is shown at the bottom of the screen; bold and light gray circles indicate available and unavailable choices, respectively. **d** Illustration of the task. Each gray circle indicates a state. Bold arrows and lines indicate participants' choices and subsequent state-transition according to the state-transition probability, respectively. Each outcome state (state 4–11) is associated with a reward (colored tokens or no token represented by a gray mosaic image). The reward probability is 0.8.

high complexity condition, respectively. To prevent state-space representations from being too complex, we limit the number of available choices to two in the first stage of each trial, while the choice availability in the second stages are either set to two or four (Fig. 1b, c). The manipulation of choice availability creates wide variability in the number of ways to achieve each goal, causing the difficulty level on each trial to range from easy to arduous. This design therefore provides four different types of conditions. Each condition is associated with a different level of computational cost (low/high × uncertainty/complexity; see Supplementary Fig. 1). Participants then make sequential choices in order to obtain different colored tokens (Fig. 1d).

Another feature of the MDP is that participants could take actions in order to obtain one of three different tokens, a silver, red, or blue token (Fig. 1c). On each trial, the relative value of the tokens, in terms of the rate of exchange of each token for real-world money (US cents), is flexibly assigned, as revealed at the beginning of each trial. For example, on a given trial, the silver token if won on that trial might yield 6 US cents, the red token, 1 US cents, and the blue token 9 cents, while the allocations could be different on the next trial. This design feature is intended to

induce trial-by-trial changes in goal values, thereby also inducing variance in RPEs and hence the reliability of MF across trials.

**Behavioral results**. Twenty-four adult participants (12 females, age between 19 and 55) performed the task, and among them 22 participants were scanned with fMRI. The task performance of participants in terms of both the total amount of earned reward and the proportion of optimal choices is significantly greater than chance level in all conditions (paired $t$-test; $p < 1e-5$).

Our task design incorporates a specific behavioral marker, a choice bias, which could indicate goal-specific planning of the model-based controller (for full details, see Supplementary Methods—choice bias and Fig. 2a). We found evidence in subjects' data to fully dismiss the possibility that subjects use a pure model-free control strategy (Fig. 2b).

To provide a direct test of the extent to which participants' behavior is under MB control, we defined an independent behavioral measure called choice optimality which quantifies the extent to which participants took the objectively best choice had they complete access to the task state-space, and a perfect ability

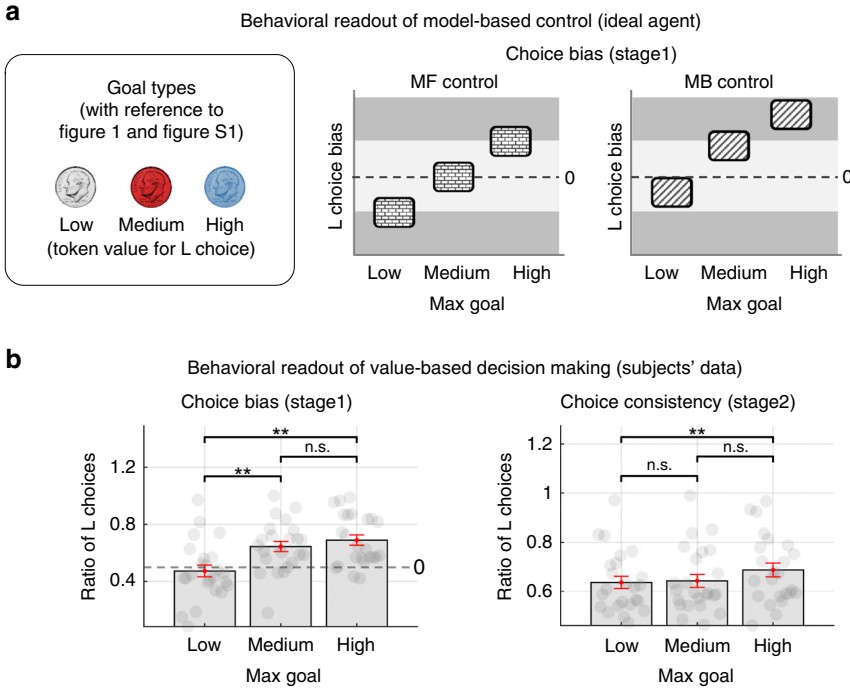

**Fig. 2 Behavioral results—choice bias and choice consistency. a** Predicted choice bias patterns of model-free (MF) and model-based (MB) control, calculated for the three goal conditions defined as the trials according to which coin has the maximum monetary outcome value (low, medium, and high token value for the L choice). Owing to the asymmetric association between outcome states and coin types (For full details, see Supplementary Methods— Choice bias), participants would exhibit distinct choice bias patterns for each goal condition that distinguishes model-based from model-free control; the MF control agent would exhibit a balanced choice bias pattern, whereas the MB control agent would show a slight left bias pattern. For full details of this measure, refer to Supplementary Methods—Behavioral measure. **b** Participants' choice bias and choice consistency, conventional behavioral markers indicating reward-based learning. Error bars are SEM across subjects. The prediction about the choice bias matches subjects' actual choice bias (the left of the below figure). In particular, the data show a clear left bias pattern, rejecting the null hypothesis that subjects used a pure model-free control strategy. This bias is also reflected in choice consistency (the right plot). These results also indicate that participants' choice behavior is guided by reward-based learning more generally.

to plan actions in that state-space. It is defined as the ratio of trials on which the subject's choice matches with the choice of an ideal agent assumed to have full access to the true state-space model (for more details, see Methods and Supplementary Methods). Choice optimality is a proxy of the extent to which participants engage in model-based control, because assuming complete knowledge of the state-space and no cognitive constraints, the MB agent will always choose more optimally than an MF agent. This prediction is confirmed by an independent computer simulation with pure MB and MF learning agents (Fig. 3a). The simulation also provides a more specific prediction that choice optimality would be greater for MB than MF strategies across all levels of uncertainty and complexity in this task (Fig. 3b).

We found a strong effect of uncertainty and complexity on choice optimality in participant's actual behavior (Fig. 3c; two-way repeated measures ANOVA; $p < 1e\text{-}4$ for both the main effect of uncertainty and the interaction effect; see Supplementary Table 3). We also found that uncertainty and complexity explain the highest variance in choice optimality when contrasting the effects of those variables against other plausible variables (Fig. 3d).

Although choice optimality provides a model independent profile of MB control, an open question is what specific patterns of choice behavior lead to high choice optimality. To address this, we focused on choice behavior after a change in token values has occurred that necessitates a change in the goal compared to the previous trial. The degree to which people switch strategy in response to a changing goal should relate to the extent to which

they are engaging MB control. Accordingly, choice switching associated with goal change (in combination with also choosing a better alternative on the next trial) provides a good account of choice optimality (Supplementary Fig. 2).

In summary, we formally established a link between the experimental manipulations (goal changes, uncertainty, and complexity) and the participants' choice behavior (choice switching), choice optimality, and learning strategies.

**Computational model incorporating uncertainty and complexity**. To test our computational hypotheses regarding the effect of uncertainty and complexity on choice behavior, we built a normative model of arbitration control[31] (see Fig. 4). A simpler version of this model was previously found to account well for arbitration between MB and MF RL in both behavioral[30,32] and fMRI data[30]. In the new version of this model, preference for MB and MF RL—$P_{MB}$ is a function of both prediction uncertainty and task complexity. Prediction uncertainty refers to estimation uncertainty about state-action-state transitions and rewards. These signals are estimated using SPE and RPE signals that underpin MB and MF learning respectively. We specifically hypothesized that task complexity also influences the transition between MB and MF RL.

Our new model becomes equivalent to our previous arbitration model[30] if task complexity is set to a low constant level as in a two-stage Markov decision task without a complexity perturbation. Further, when the environment is perfectly stable (i.e., a fixed amount of state-transition uncertainty and a fixed level of

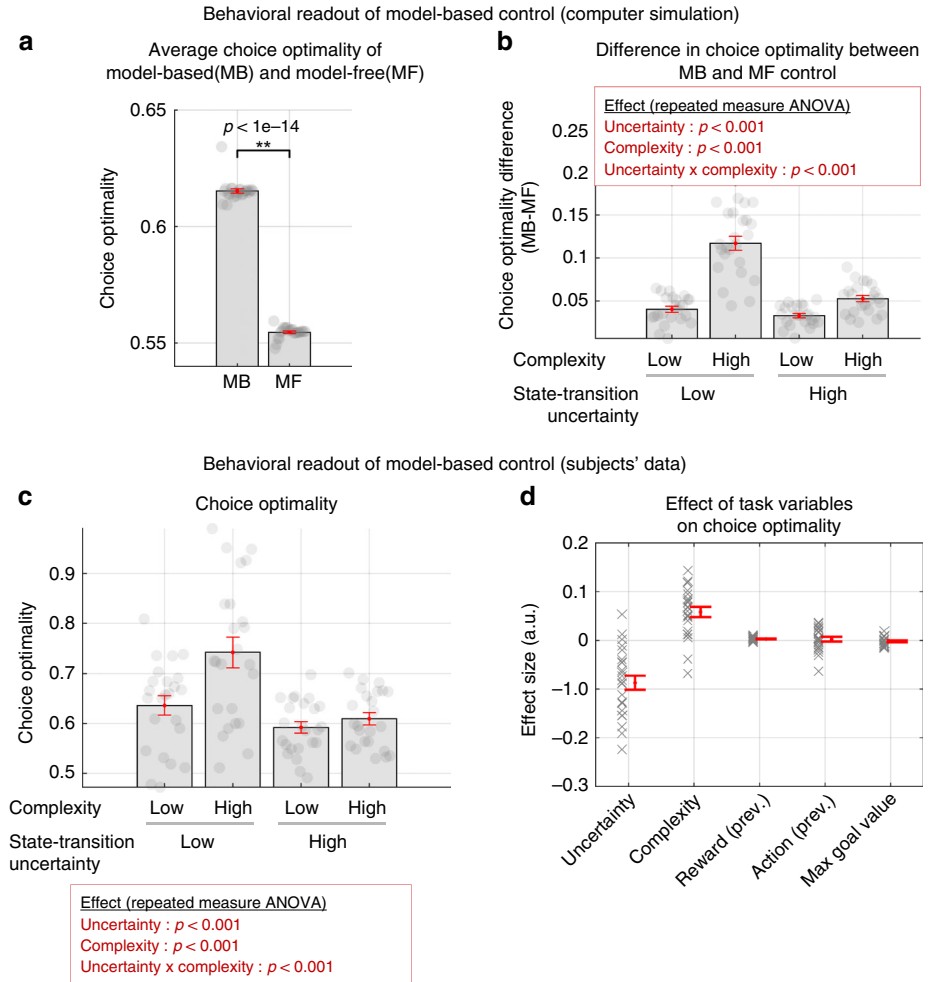

**Fig. 3 Behavioral results— choice optimality. a** Choice optimality (a proxy for assessing the degree of agents' engagement in model-based control) of a model-based and model-free RL agent. Each MB and MF model simulation (that generates each one of the data points) was produced using free parameters derived from separate fits of each of these models to each individual participant's behavioral data in the study. Choice optimality depicts the degree of match between agents' actual choices and an ideal agent's choice corrected for the number of available options. For full details of this measure, refer to Methods. **b** Difference in choice optimality between an MB and MF agent for the four experimental conditions (low/high state-transition uncertainty × low/high-task complexity). Shown in red boxes are the effect of the two experimental variables on each measure (two-way repeated measures ANOVA). **c** Participants' choice optimality for the four experimental conditions. Shown in red boxes are the effect of the two experimental variables on each measure (two-way repeated measures ANOVA; also see Supplementary Table 3 for full details). **d** Results of a general linear model analysis (dependent variable: choice optimality, independent variables: uncertainty, complexity, reward values, choices in the previous trial, and goal values). Uncertainty and complexity, the two key experimental variables in our task, significantly influence choice optimality (paired $t$-test; $p < 0.001$). Error bars are SEM across subjects.

task complexity), the particulars of this model converge to a stable mixture of MB and MF RL[7].

The process of our computational model is described as follows: first, in response to the agent's action on each trial, the environment provides the model with the state-action-state transition, token values, and task complexity. These observations are then used to compute the transition rates (MB→MF and MF→MB), which subsequently determines the model choice probability $P_{MB}$. Second, the model integrates MB and MF value estimations to compute an overall integrated action value ($Q(s,a)$ of Fig. 4), which is subsequently translated into an action ($P(a|s)$ of Fig. 4). It is noted that we use this framework to formally implement various hypotheses about the effect of uncertainty and complexity on RL. For instance, the configuration of the model that best accounts for subjects' choice behavior would specify the way people combine MB and MF RL to tailor their behavior to account for the degree of uncertainty and complexity of the environment.

**Effects of uncertainty and complexity on learning**. To determine how task complexity is embedded into the arbitration control process, we formulated a variety of possible model implementations, which we could then fully permute and test in a large-scale model comparison, described briefly as follows (see Fig. 5a and Methods section for more details of model specification):

The first factor in the hypothesis set is the effect of complexity on arbitration control. We considered different ways in which complexity could impact on the allocation of MB vs MF control, including whether the modulation was excitatory or inhibitory, i.e. whether complexity influence on the arbitration process positively or negatively. Also considered was the direction of modulation: does complexity effect the transition of behavioral control from model-free to model-based, from model-based to model-free, or does the transition operate in both directions (see Methods). We also considered several ways in which the effects of uncertainty might interact with the effects of complexity (see Methods).

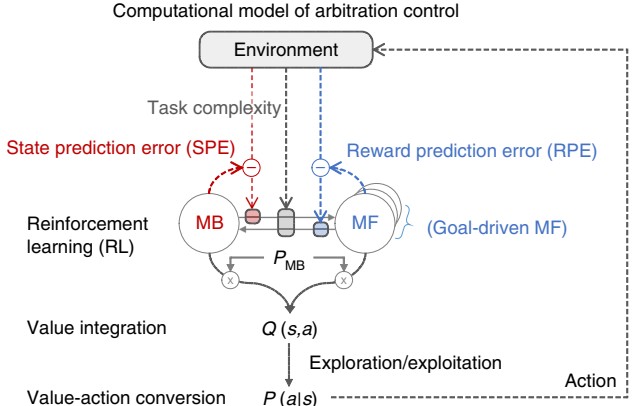

Computational model of arbitration control

**Fig. 4 Computational model of arbitration control incorporating uncertainty and complexity.** The circle-and-arrow illustration depicts a two-state dynamic transition model, in which the current state depends on the previous state (an endogenous variable) and input from the environment (exogenous variables). The environmental input includes the state-transition which elicits state-prediction errors (SPEs), rewards that elicit reward-prediction error (RPEs), and the task complexity. The arrow refers to the transition rate from MB to MF RL or vice versa, which is a function of SPE, RPE, and task complexity. The circle refers to the state, defined as the probability of choosing MB RL ($P_{MB}$). $Q(s,a)$ refers to the values of the currently available actions ($a$) in the current state ($s$). The value is then translated into action, indicated by the action choice probability $P(a|s)$.

The second factor is the effect of complexity on the choice itself. We tested whether task complexity impacted on the soft-max choice temperature that sets the stochasticity of the choices of the integrated model, by comparing the case in which there is no effect of complexity on the choice temperature parameter (Null), a case in which increasing complexity increases the degree of explorative choices, and the case where increasing complexity decreases the degree of choice exploration.

Thirdly, we considered the implementation of the model-free algorithm. Recall that at the beginning of each trial, the participant is presented with the current value of each of the three tokens. The most naïve implementation of model-free RL would ignore those token values and treat each trial the same irrespective of what values are assigned to particular tokens. Alternatively and more plausibly, token values could be embedded into the state space itself, on which the model-free agent learns. For this possibility, we built a modified model-free algorithm that divided up the task into three unique state-space representations depending on which token was the dominant goal (which depends on which token was the most valuable on a given trial), a model variant we call the 3Q model. Another, perhaps less plausible possibility is that three completely separate model-free strategies exist to learn about the separate values of each possible goal, which we call the 3MF model. Thus, we tested three classes of model-free agent, ranging from a simple model-free agent that does not differentiate between token states, a model that treats the most valuable token as identifying one of three relevant subsets of the state-space (red token goal, blue token goal, and silver token goal) and a model that assumes three independent model-free agents for each selected goal.

The effect of uncertainty on the transition rate was implemented as in our previous model[30]. Each of these factors (and sub-factors) was fully tested in each possible combination with each other factor, rendering a total of 117 model variants, that also included as a baseline model, the original arbitration scheme from our 2014 paper that only incorporated uncertainty as a

variable and not complexity (see Fig. 4). We then compared the extent to which each of those models could explain the behavioral data using across all of these models. For this, we fit each version of the model to each individual subject's data, and then ran a Bayesian model selection[33], with exceedance probability $p > 0.99$ (Fig. 5; see Model comparison in Supplementary Methods).

We found that one specific model variant provided a dominant account of the behavioral data, with an exceedance probability of 0.99 (Information about model parameter values are provided in Supplementary Table 1 and Supplementary Fig. 3, respectively). The exceedance probability for the best-fitting model was still 0.97 when restricting our model comparison process to only the best-fitting 5 models from the original large-scale BMS analysis, suggesting that these findings are not an artifact of running a large-scale model comparison (Supplementary Fig. 4). In the best-fitting model, an increase in task complexity exerted a positive modulatory effect on the transition between MF and MB control. That is, the best-fitting model supported an effect of complexity on arbitration such that an increase in complexity produced an increased tendency to transition from MF to MB control. Recall, that this is not compatible with our initial hypothesis that increased complexity would generally tax the accuracy of the MB controller, thereby resulting in an increase in MF control. However, the best-fitting model also prescribed that uncertainty and complexity interact, such that under conditions of both high uncertainty AND high complexity, MF control would become favored. We will describe in more detail the nature of this interaction in the section below. Secondly, the best-fitting model had the feature that increasing complexity increases the degree of exploration, suggesting that subjects tend to explore more under conditions of high-task complexity. Finally, the best-fitting model variant also had the feature that the state-space for the MF agent was sub-divided according to which goal was currently selected (assuming that the goal selected corresponded to the maximum token value), i.e. the 3Q model, as opposed to a single MF agent that ignores token values, or separate MF agents. In summary, we found evidence for a model incorporating the effects of both uncertainty and complexity in arbitration between MB and MF RL, and that these two variables interact to drive arbitration as detailed in the following section.

**Effect of uncertainty and complexity on arbitration control.** Choice behavior generated by the best-fitting model also describes the extent to which participants' actual choice behavior can be shown to be guided by reward-based learning more generally (Fig. 6a). In addition, we ran a parameter recovery analysis to further establish a link between choice behavior and the computations underlying arbitration control, and found that the model's key parameters were successfully recovered from the behavior of the best-fitting model (Parameter recovery analysis; Supplementary Fig. 5).

We also attempted to get a direct behavioral readout of model-based control for the four blocked experimental conditions: low/high uncertainty × low/high complexity. Specifically, we checked whether the effect of uncertainty and complexity on choice optimality also exists in the model's predicted behavior. Notably, when the best-fitting model performed the task, we found that human participants' actual choice optimality patterns (Fig. 3c) are predicted well by our computational model (Fig. 6b).

To gain more insight into the role of uncertainty and complexity on choice of the MB vs MF RL strategy, we assessed the degree of engagement of the model-based control of our computational model. For this we examined the model weight $P_{MB}$. The $P_{MB}$ weight was binned and averaged within each subject according to whether or not the trial was high or low in

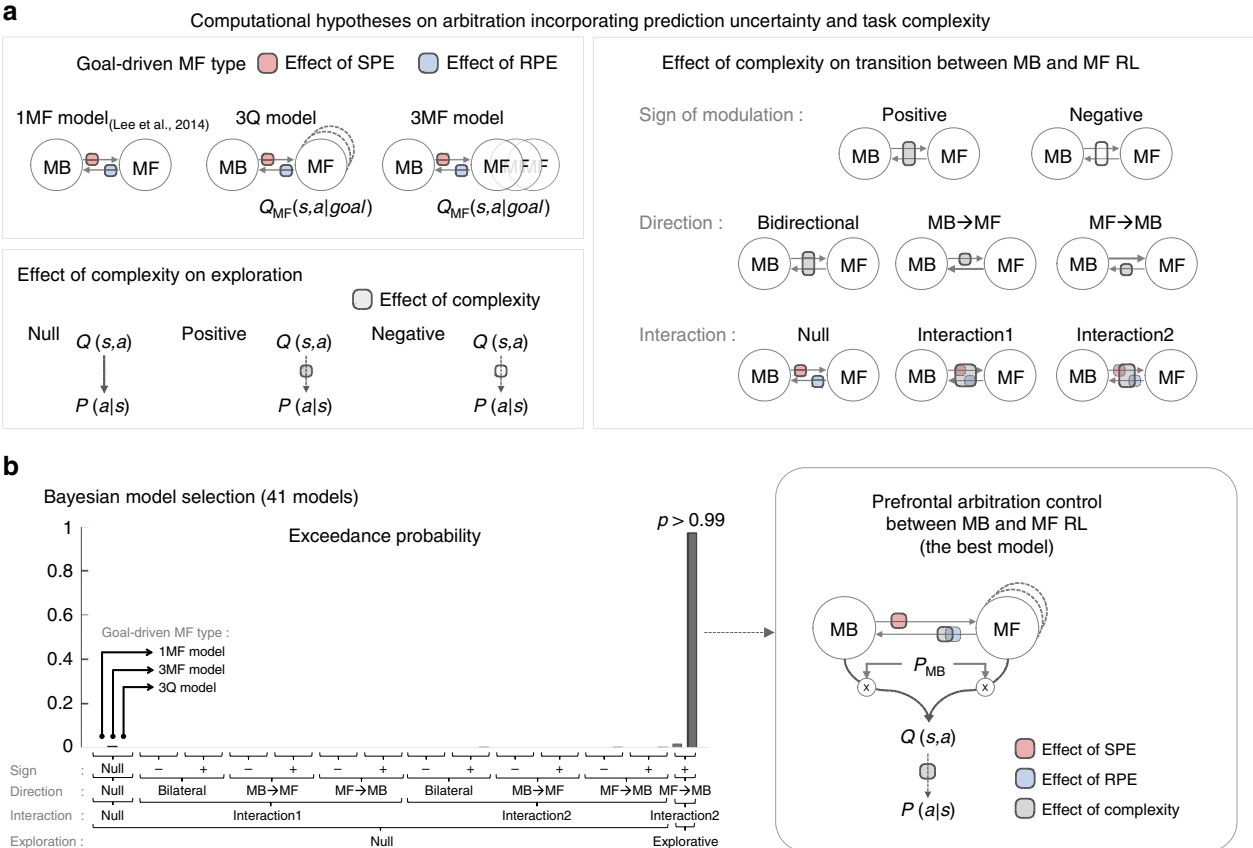

**Fig. 5** Model comparison analysis on behavioral data. **a** We ran a large-scale Bayesian model selection analysis to compare different versions of arbitration control. These model variants were broadly classified as reflecting the effect of complexity on the transition between MB and MF RL ($13 = 1 + 2 \times 2 \times 3$ types), the effect of complexity on exploration (3 variants), and the form of the MF controller (3 variants) each of which is classified by a type of goal-driven MF (3 types), an effect of complexity on transition between MB and MF RL, and an effect of complexity on exploration (3 types). Lee2014 refers to the original arbitration model[30]. **b** Results of the Bayesian model selection analysis. Among a total of 117 versions, we show only 41 major cases for simplicity, including the original arbitration model and 40 other different versions that show non-trivial performance (but the same result holds if running the full model comparison across the 117 versions). The model that best accounts for behavior is the version {3Q model, interaction type2, excitatory modulation on MF→MB, explorative} (exceedance probability >0.99; model parameter values and distributions are shown in Supplementary Table 1 and Supplementary Fig. 3, respectively).

complexity and high or low in state-space uncertainty, and the fitted model-weights were then averaged across participants. Note these are model fits, illustrative of model performance as fit to the behavioral data, rather than being directly informative about participants' actual behavior. In essence, this is a way to understand the behavioral predictions of the model itself. When interrogating the fitted model, we found an effect of uncertainty and complexity on the weighting between MB and MF control (Fig. 6c; two-way repeated measures ANOVA; $p < 1e-4$ for the main effect of both state-transition uncertainty and task complexity; $p = 0.039$ for the interaction effect; full statistics are shown in Supplementary Table 4). Specifically, according to the model, MB control is preferred when the degree of task complexity increases, whereas MF is favored when the amount of state-space uncertainty increases. A more intriguing finding is that the increase in state-transition uncertainty tends to nullify the effect of task complexity or vice versa.

To further compare the degree of influence of uncertainty and complexity on choice optimality within the model, we ran a general linear model (GLM) analysis on the model's behavioral data, in which uncertainty and complexity were regressed against choice optimality, and we ran another GLM analysis on the actual participants' behavioral data. The model's behavioral data were generated by running simulations with our model on the

behavioral task. We found a significant correlation between the effect sizes of these two cases (Fig. 6d, e), suggesting that our model encapsulates the essence of participants' behavior as guided by an arbitration determined mix of MB and MF RL. It is noted that, our model, which incorporates both uncertainty and complexity, accounts for behavior significantly better than the original Lee2014 model that incorporates only uncertainty (refer to Supplementary Fig. 6).

In summary, these results provide both a computational and behavioral account of how participants regulate the tradeoff between MB and MF RL in the presence of uncertainty and task complexity: they tend to favor use of a MB RL strategy under conditions of high compared to low task complexity, while at the same time they tend to resort to MF RL when the amount of state-space uncertainty increases to the level at which the MB RL strategy can no longer provide reliable predictions. However, these variables interact such that under conditions of both high complexity and high uncertainty, model-free control is favored over and above the effects of each of these two variables alone.

**Neural representations of model-based and model-free RL.** To provide a neural account of MB and MF RL, we ran a GLM analysis on the fMRI data in which each variable of the

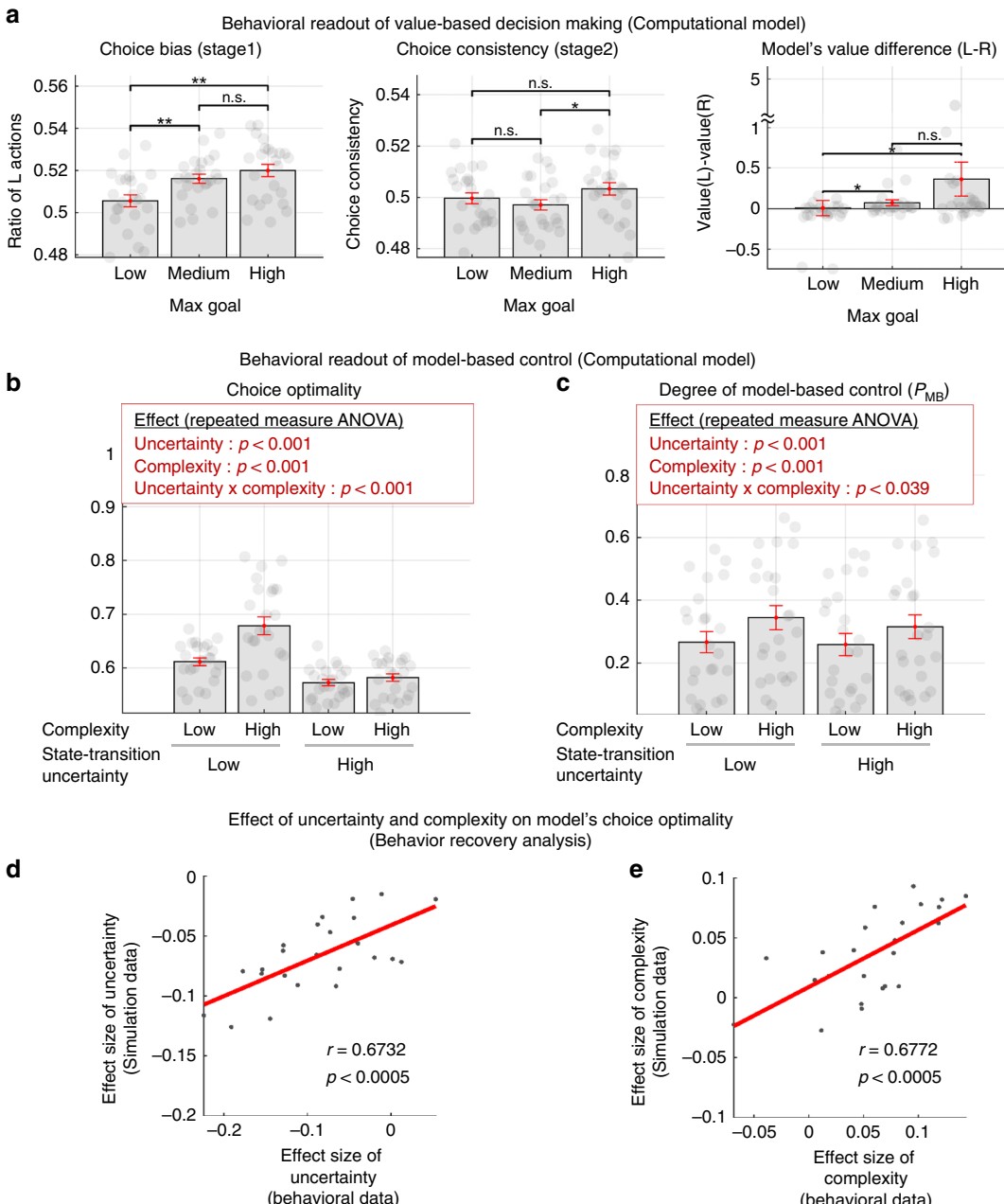

**Fig. 6 Computational model fitting results. a** Choice bias (left), choice consistency (middle), and the average value difference (right) of our computational model of arbitration control (Fig. 5b). For this, we ran a deterministic simulation in which the best-fitting version of the arbitration model, using parameters obtained from fitting to participants behavior, experiences exactly the same episode of events as each individual subject, and we generated the trial-by-trial outputs. The max goal conditions are defined in the same way as in the Fig. 2a. Error bars are SEM across subjects. Note that both the choice bias and choice consistency patterns of the model (the left and the middle plot) are fully consistent with the behavioral results (Fig. 2b). Second, the values difference (left–right choice) of the model is also consistent with this finding (the right plot), suggesting that these behavioral patterns are originated from value learning. In summary, our computational model encapsulates the essence of subjects' choice behavior guided by reward-based learning. **b** Patterns of choice optimality generated by the best-fitting model, using parameters obtained from fitting to participants behavior. For this, the model was run on the task (1000 times), and we computed choice optimality measures in the same way as in Fig. 3. **c** Degree of engagement of model-based control predicted by the computational model, based on the model fits to individual participants. $P_{MB}$ corresponds to the weights allocated to the MB strategy. Shown in the red box are the effect of the two experimental variables on each measure (two-way repeated measures ANOVA; also see Supplementary Table 4 for full details). Error bars are SEM across subjects. **d**, **e** Behavioral effect recovery analysis. The individual effect sizes of uncertainty (**d**) and complexity (**e**) on choice optimality of subjects (true data) were compared with those of our computational model (simulated data).

computational model that best-fit behavior is regressed against the fMRI data (see Methods). First, we replicated previous findings indicating neural encoding of prediction error signals, SPE and RPE, two key variables necessary for updating MB and MF RL values (see Supplementary Table 2). Consistent with previous

findings, we found SPE signals in dorsolateral prefrontal cortex (dlPFC) ($p < 0.05$ family-wise error (FWE) corrected)[24,30] and RPE signals in the ventral striatum ($p < 0.05$ small volume corrected)[30,34,35]. Second, we replicated chosen value signal correlates for the MB and MF controllers. The MB value signal was

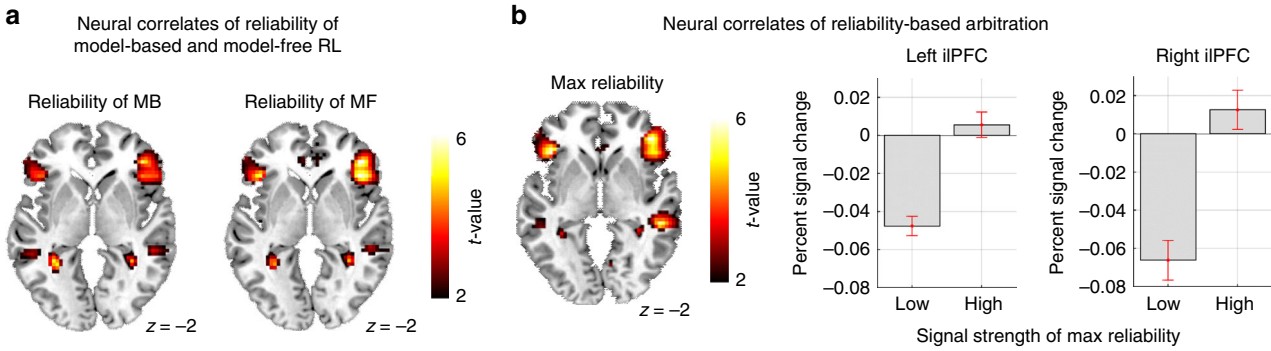

**Fig. 7 Neural signatures of model-free and model-based RL and arbitration control. a** Bilateral ilPFC encodes reliability signals for both the MB and the MF systems. Note that the two signals are not highly correlated (absolute mean correlation <0.3); this task design was previously shown to successfully dissociate the two types of RL[30]. Threshold is set at $p < 0.005$. **b** (Left) Inferior lateral prefrontal cortex bilaterally encodes reliability information on each trial of both MB and MF RL, as well as whichever strategy that provides more accurate predictions ("max reliability"[30]). (Right) The mean percent signal change for a parametric modulator encoding a max reliability signal in the inferior lateral prefrontal cortex (ilPFC). The signal has been split into two equal-sized bins according to the 50th and 100th percentile. The error bars are SEM across subjects.

associated with BOLD activity in multiple areas within the PFC ($p < 0.05$ cluster-level corrected), whereas the MF value signal was found in supplementary motor area (SMA) ($p < 0.05$ FWE corrected) and notably posterior putamen (significantly at $p < 0.05$ small volume corrected), which has previously been implicated in MF valuation[10,36]. Third, we found evidence in the ventromedial prefrontal cortex (vmPFC) for an integrated value signal that combines model-based and model-free value predictions according to their weighted combination as determined by the arbitration process ($Q(s,a)$ shown in Supplementary Table 2; significantly at $p < 0.05$ small volume corrected). In summary, we replicated existing findings about fMRI correlates of variables necessary to implement MB and MR RL.

In addition, we found additional evidence for the implementation of the goal-driven MF model (Supplementary Table 2; the definition of the regressor is provided in Supplementary Methods), which is that the activity of medial frontal gyrus was found to be bilaterally correlated with the goal change signals ($p < 0.05$ FWE corrected; Supplementary Fig. 7), information necessary for the agent to cache out an MF value signal in order to achieve a new goal.

**Arbitration signals in prefrontal cortex**. We next examined the fMRI data for arbitration signals. Replicating our previous results[30], we found correlates of reliability signals for both MB and MF controllers in the inferior lateral prefrontal cortex (ilPFC) bilaterally (Fig. 7a). But the activity of ilPFC was most strongly associated with the maximum of the reliability of the MB and MF systems, that is, when using a regressor in which the reliability value of whichever system was most reliable on a given trial is input as the value for that trial (Fig. 7b; $p < 0.05$ cluster-level corrected). These findings are again successful replications of findings from our previous study[30]. Note that the activity of ilPFC also reflects an alternative (not based on our computational fMRI analysis) measure of model-based control, choice optimality (Supplementary Fig. 8).

**Model comparison against fMRI data**. Next to formally test our main hypothesis of uncertainty and complexity-sensitive arbitration control, we compared two separate arbitration models against the fMRI data. One was the best-fitting model described above in which both task complexity and reliability are taken into account as playing a role in driving the arbitration process. The second, was a model in which only reliability was involved in the

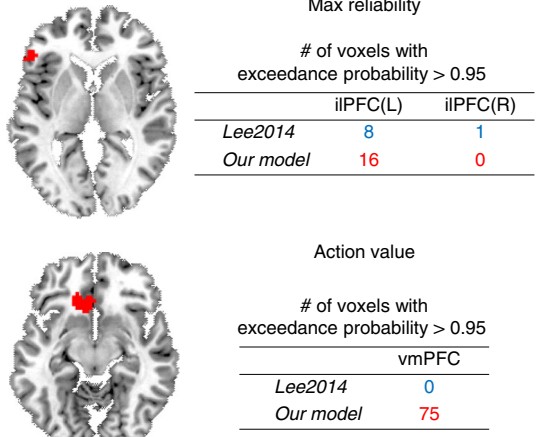

**Fig. 8 Results of a Bayesian model selection analysis.** The red blobs and table show the voxels and the number of voxels, respectively, that favor each model with an exceedance probability >0.95, indicating that the corresponding model provides a significantly better account for the BOLD activity in that region. Lee2014 refers to an arbitration control that takes into account only uncertainty as used by Lee et al.[30]. Current model refers to the arbitration control model that was selected in the model comparison based on the behavioral data which incorporates both prediction uncertainty and task complexity. For an unbiased test, the coordinates of the ilPFC and the vmPFC ROIs were taken from ref. [30].

arbitration process[30]. We then compared the fit of these two models to the fMRI data in two brain regions, ilPFC for reliability signals and vmPFC for valuation signals. For this we ran a Bayesian model selection analysis[33], using spherical ROIs centered on the coordinates from our 2014 study, thereby ensuring independence of the ROI selection from the current dataset. In a majority of voxels in both regions, reliability signals from the model incorporating both reliability and task complexity were preferred over the previous model incorporating reliability only (Fig. 8). These findings go beyond our original 2014 findings by providing evidence that the model in which complexity is taken into account provides a better account of prefrontal-mediated arbitration control.

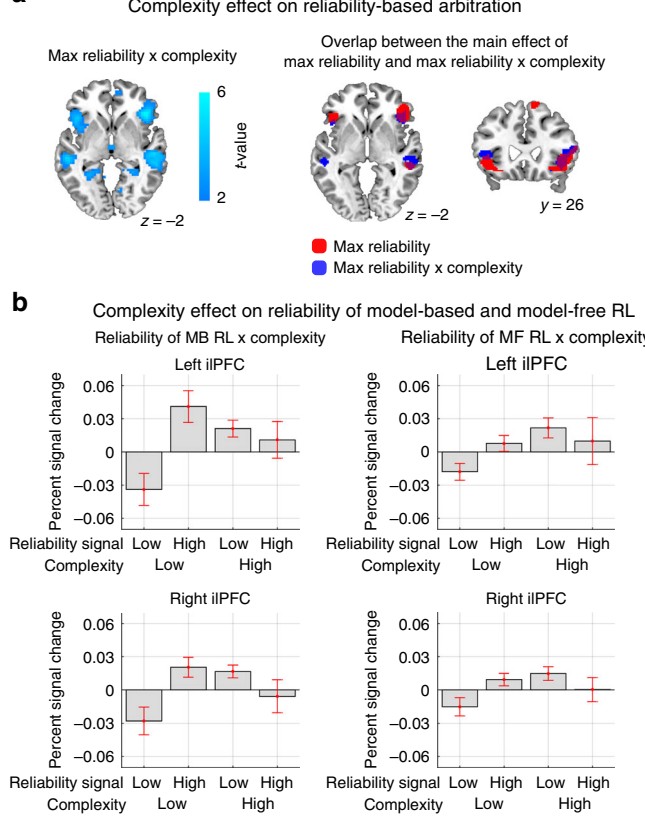

**Fig. 9 Modulation of inferior prefrontal reliability by complexity. a** (Left) Bilateral ilPFC was found to exhibit a significant interaction between complexity and reliability (Max reliability × complexity). Statistical significance of the negative effects is illustrated by the cyan colormap. The threshold is set at $p < 0.005$. (Right) The brain region reflecting the interaction effect largely overlaps with the brain area implicated in arbitration control. The red and blue regions refers to the main effect of max reliability and the interaction between reliability and task complexity, respectively, thresholded at $p < 0.001$. **b** Plot of average signal change extracted from left and right ilPFC clusters showing the interaction, shown separately for reliability signals derived from the MF and MB controllers. Data are split into two equal-sized bins according to the 50th and 100th percentile of the reliability signal, and shown for the trials in the low and high complexity condition separately. The error bars are SEM across subjects.

**Modulation of inferior prefrontal reliability by complexity.** We have thus far demonstrated that reliability signals from an arbitration model incorporating task complexity provides a better account of fMRI activity than reliability signals derived from a model that does not incorporate complexity. To test for an explicit contribution of task complexity to the neural arbitration signal, we next ran an additional fMRI analysis in which we included a parametric regressor denoting the onset of trials of high vs low task complexity as a separate regressor of interest. We then entered another additional regressor, corresponding to the formal interaction of Max reliability with task complexity to reveal areas in which the reliability signal is modulated differently depending on whether a specific trial is high or low in complexity. While we found no significant effect of the main effect of task complexity in our main regions of interest (Supplementary Table 2), we found evidence for a significant interaction effect of complexity and reliability. A region of ilPFC bilaterally was found to show a significant negative interaction between complexity and reliability (Fig. 9a). This region was found to overlap

substantively with the regions of ilPFC found to exhibit a main effect of reliability (Fig. 9b). To visualize the interaction effect, in a post-hoc analysis we extracted the average % signal change from clusters exhibiting the interaction in left and right ilPFC respectively. We binned the signal according to whether reliability was high or low, and whether complexity was high or low, shown in Fig. 9b. Reliability signals are plotted separately for MF and MB reliability, although the results are similar for max reliability. As can be seen, the reliability signals show evidence of being attenuated particularly when complexity is high relative to when complexity is low. This shows how these two arbitration signals interact in ilPFC.

## Discussion

We provide evidence supporting the interaction of two key variables in arbitration control between MB and MF RL. In addition to replicating our previous finding implicating the uncertainty (or reliability) of the predictions made by the model-based and model-free controllers in moderating the influence of these two systems over behavior, we found evidence that state-space complexity also contributes to setting the balance between these two systems. These behavioral results were supported by evidence that a region of the brain previously implicated in the arbitration process, the ilPFC not only encodes signals related to the reliability of the predictions of the two systems that would support an uncertainty-based arbitration mechanism, but furthermore that activity in this region is better accounted for by an arbitration model that also incorporates the effects of task complexity into the arbitration process. Moreover, we found evidence that task complexity and reliability appear to directly interact in this region. Taken together, these findings help advance our understanding of the contribution of two key variables to the arbitration process at behavioral and neural levels.

We found direct evidence for a contribution of task complexity to arbitration. In our large-scale model comparison we found empirical support for a version of the arbitration process in which the complexity variable has a positive modulation effect on the transitions from MF to MB. Second, this is corroborated by the fact that the best-fitting model exhibited an increased preference for MB over MF in the high complexity condition on average. Third, in an independent behavioral analysis which uses choice optimality to quantify the extent to which choice behavior is guided by MB RL, we found that subjects' choice optimality increases with the degree of task complexity. These results together suggest that an increase in task complexity creates an overall bias towards MB RL, contrary to our initial hypothesis in which we considered that increased complexity would tax the model-based system resulting in increased model-free control. Another interesting finding supporting this idea is that an increase in task complexity makes choices more flexible and explorative. In summary, these findings suggest that humans attempt to resolve task complexity by engaging a more explorative MB RL strategy.

We also found an effect of state-space uncertainty on the arbitration process. Specifically, very high state-space uncertainty makes participants resort more to an MF RL strategy. This effect arises because high state-space uncertainty results in a lowered reliability of the predictions of the model-based controller, thereby resulting in a reduced contribution of behavior of the model-based controller. It should be noted that, the model-based controller should generally compute a more accurate prediction than its model-free counterpart, which by contrast necessarily generates approximate value predictions. However, this holds only under the situation where the model-based controller has access to a reliable model of the state-space. If its state-space model is not reliable or accurate, then the model-based controller

cannot generate accurate predictions about the value of different actions. In this task, we influence the extent to which the model-based controller has access to reliable predictions about the state-space by directly modulating the state-space uncertainty. Thus, under conditions in which the state-space model is highly unreliable, humans appear to rely more on model-free control. Conversely, as we have shown previously[30], if the reliability of the model-free controller is decreased, participants will all else being equal rely more on model-based control. Thus, uncertainty in the predictions of these controllers appears to play a key role in underpinning the arbitration process between them.

In addition to the main effects of complexity and state-space uncertainty, we have shown that these two variables interact. Under conditions where both state-space uncertainty is high and complexity is high, the MB system appears to be disproportionately affected, in that participants abandon MB-based control in favor of MF control. Thus, our hypothesis about an effect of complexity resulting in decreased MB control was borne out in a qualified manner, in that this effect only happens when state-space uncertainty is high. This finding suggests that the arbitration process takes into account the effects of both of these variables at the same time, and dynamically finds a tradeoff between them. Participants appear to use MB RL to resolve uncertainty and complexity, but owing to the fact that MB RL is more cognitively demanding than MF RL, they resort to the default strategy, MF RL, when the performance gain of MB RL does not outweigh the level of cognitive load required for MB RL. Another way of interpreting these findings is that when task demands increase but the MB system is capable of meeting those challenges, then MB control can and does step up to meet the challenge, but if task demands get beyond the capacity of the MB system, then MF control takes over by default. It is likely that individual differences in executive function such as working memory capacity will moderate this effect across participants, as has been shown in the case of other challenges to MB control such as stress induction[37,38].

The present findings are also relevant to the predictions of expected value of control (EVC) theory[39,40]. While the theory predicts that increasing task difficulty brings about an increase in the intensity of cognitive control signals, the theory itself does not offer a direct prediction about how control signal intensity influences RL. Our computational model explains how the brain chooses between MB and MF RL with varying degrees of cognitive control intensity, and furthermore, why this choice is made.

The model comparison analysis of the present study also revealed that task complexity affects transitions from MF to MB RL, but not transitions in the other direction. This finding provides further evidence to support the existence of an asymmetry in arbitration control such that arbitration is performed in a way that selectively gates the MF system[10,30]. These results may be reasonable from an evolutionary perspective in that the implementation of MF learning in parts of the basal ganglia may have arisen earlier on in the evolutionary history of adaptive intelligence, while later on, cortically mediated MB control may have emerged so as to deal with more complex situations.

Our study also advances understanding of inferior prefrontal cortex computations during arbitration. This region was found to encode not only the reliability of the two systems[30] but also task complexity. The reliability signal itself was found to reflect the effects of task complexity, as shown by the formal model comparison in which activity in this region was better accounted for by an arbitration model that incorporated task complexity compared one ignoring task complexity. Moreover, we found that when testing directly for an interaction between the reliability signals and complexity, an overlapping region of inferior prefrontal cortex showed evidence for a significant interaction

between these signals. These findings demonstrate that task complexity directly modulates the putative neural correlates of the arbitration process.

It is also important to acknowledge that a number of open questions remain. A fundamental question concerns how arbitration computations within inferior prefrontal cortex are actually implemented at the neuronal level. While our findings show BOLD responses related to various arbitration related-signals in this region, how these signals are utilized at the neural level to implement the arbitration process is not known. Building on the present study and earlier studies investigating executive control mechanisms in prefrontal-striatal circuitry[7,8,23,41–51], more biologically plausible models of the arbitration process could be developed to go beyond the algorithmic level used in the present study. Furthermore, to guide the development of such models, a better understanding of the underlying neuronal dynamics in these prefrontal regions during the arbitration process is necessary, suggesting the utility of techniques with better spatial and temporal resolution than fMRI.

An important limitation of the task used here is that behavior is studied under conditions of high instability and/or variability in transitions between MB and MF control. This is done by design, because to maximize signal detectability within the framework of an fMRI study, we needed to maximize the variance in the transition between these two different forms of control. However, in real-world behavior, it could be expected that transitions between MB and MF control would typically evolve at a slower pace. One of the main advantages of MF control is the lower computational cost entailed by engaging cached values learned via model-free RL compared to model-based RL. However, in the long run the MB system should cease computing action-values when the MF system is in control, as otherwise the cost advantage gained by increasing MF control would be moot. A limitation of the present model is that it assumes that MB values continue to be computed throughout the task. This is so because we did not find behavioral evidence that such signals ceased to be estimated during the task, which would be manifested at the behavioral level by complete dominance of MF control over behavior. However, we suspect that in real-world behavior, the MB system would eventually go offline, and this should be reflected in behavioral dominance of MF control. More generally, it will be important to study the behavior of MB and MF controllers across a wide range of tasks and experimental conditions to gain a more complete understanding of the arbitration process.

Finally, the task complexity manipulation used here by which we increased the number of available state-spaces available in the decision problem can also impact on the MF system, because the increase in the number of actions to be learned means that the MF system has less opportunity to sample those state-action pairs, thereby having less opportunity to acquire accurate value representations. Thus, the trade-off under these conditions is more complicated than the effects of computational cost on MB control. Further, though we focused on complexity and uncertainty as one potential way to manipulate computational cost, it is possible our findings will not generalize to other forms of computational cost. Future studies could therefore focus on further delineating the effects of state-space complexity from other factors related to computational cost.

In conclusion, our findings provide insight into how the brain dynamically combines different RL strategies to deal with uncertainty and complexity. Our findings suggest that both of these variables are taken into account in the arbitration between MB and MF RL. Moreover, we found that such an arbitration control principle is best reflected in neural activity patterns in the ilPFC, the same area we previously found to play a pivotal role in arbitration control, thereby fostering a deeper appreciation of the role of ilPFC in arbitration control.

## Methods

**Participants.** Twenty-four right-handed volunteers (ten females, with an age range between 19 and 55) participated in the study, 22 of whom were scanned with fMRI. They were screened prior to the experiment to exclude those with a history of neurological or psychiatric illness. All participants gave informed consent, and the study was approved by the Institutional Review Board of the California Institute of Technology.

**Stimuli.** The image set for the experiment consisted of 126 fractal images to represent states, three kinds of color coins (red, blue, and silver), and four kinds of fractal images to represent outcome states associated with each color coin (red, blue, silver, and none). The colors of the outcome state image were accompanied by numerical amounts which indicate the amount of money that subjects could receive in that state. Before the experiment began, the stimulus computer randomly chose a subset of 11 fractal images to be subsequently used to represent each state in that specific participant.

**Task.** Participants performed a sequential two-choice Markov decision task, in which they need to make two sequential choices (by pressing one of four buttons: L1, L2, R1, R2) to obtain a monetary outcome (token) at the end stage. Making no choice in 4 s had a computer make a random choice to proceed and that trial was marked as a penalizing trial. Each trial begins with a presentation of values of each token in that trial, followed by a presentation a fractal image representing a starting state. The presentation of each state is accompanied by choice availability information shown at the bottom of the screen. Only two choices (L1 and R1) are available in the starting state (S1). The starting state is the same across all trials. Making a choice in the starting state is followed by a presentation of another fractal image representing one of ten states (S2–S11). The states were intersected by a variable temporal interval drawn from a uniform distribution between 1 and 4 s. The inter-trial interval was also sampled from a uniform distribution between 1 and 4 s. The reward was displayed for 2 s. At the beginning of the experiment, subjects were informed that they need to learn about the states and corresponding outcomes to collect as many coins as possible and that they will get to keep the money they cumulatively earned at the end of the experiment. Participants were not informed about the specific state-transition probabilities used in the task except they were told that the contingencies might change during the course of the experiment. In the pre-training session, to give participants an opportunity to learn about the task structure, they were given 100 trials in which they can freely navigate the state space by making any choices. During this session, the state-transition probability was fixed at (0.5, 0.5) and the values of all color tokens are fixed at 5, indicating that any token color would yield the same amount of monetary reward. The experiment proceeded in five separate scanning sessions of 80 trials each on average.

In order to effectively dissociate the model-free strategy from the model-based, the experimental design of the present study introduces two task parameters: specific goal-condition and state transition probabilities. First, to create a situation in which the model-based control is preferred over the model-free control, the present experimental design introduced a generalized version of the specific goal condition[30], in which all token values are randomly drawn from a uniform probability distribution U(1,10) from trial-to-trial. If participants reached the outcome state associated with a token, then they would gain the corresponding monetary amount. Note that this goal-value manipulation is intended to encourage participants to act on a stable model-based control strategy, as opposed to developing separate multiple model-free strategies for each color tokens in the absence of the model-based control.

Second, to create a situation where the model-free control overrides the model-based control and to further dissociate the model-free from the model-based, changes to the state transition probabilities were implemented. Two types of state-transition probability were used—(0.9, 0.1) and (0.5, 0.5) (a low and a high state-transition uncertainty condition, respectively). They are the probabilities that the choice is followed by going into the two consecutive states. For example, if you make a left choice at the state 1 and the state transition probability is (0.9, 0.1) at that moment, then the probability of your next state being state 2 is 0.9 and the probability for state 3 is 0.1. The order of the block conditions was randomized. The blocks with the state transition probability (0.9,0.1) consists of three to five trials, whereas those with (0.5,0.5) consists of five to seven trials; it was previously shown that with (0.9, 0.1) participants feel that the state transition is congruent with the choice, whereas with (0.5, 0.5) the state transition is random[30]. Furthermore, the changes at these rates ensures that tonically varying changes in model-based vs model-free control can be detected at experimental frequencies appropriate for fMRI data. The state-transition probability value was not informed to participants; estimation of state-transition probabilities can be made by using the model-based strategy.

To manipulate the state-space complexity, the present study also introduced the third task parameter, the number of available choices. Two types of choice sets were used—(L, R) and (L1, L2, R1, R2) (a low and a high state-space complexity condition, respectively). The order of the block conditions was randomized. To preclude the task being too complex, changes in the number of available choices occur only in the second stage of each trial, while in the first stage the number is always limited to two (L and R).

**Behavioral measure (choice optimality).** Choice optimality measure quantifies the extent to which participants on a given trial took the objectively best choice had they complete access to the task state-space, and a perfect ability to plan actions in that state-space. It is based on the choice of the ideal agent assumed to have a full, immediate access to information of the environmental structure, including state-transition uncertainty and task complexity. The choice optimality is defined as the degree of match between subjects' actual choices and an ideal agent's choice corrected for the number of available options. To compute the degree of choice match between the subject and the ideal agent, for each condition, we calculated an average of normalized values (i.e., likelihood) of the ideal agent for the choice that a subject actually made on each trial. To correct for the number of options, we then multiplied it by 2 for the high complexity condition; this is intended to compensate for the effect that the baseline level of the likelihood in the high complexity condition (# of available options = 4) becomes half of that in the low complexity condition (# of available options = 2). In other words, this adjustment effectively compensates the effect of # of available options on normalization without biasing the correspondence between participant's choices and optimal choices. The choice optimality value would have a maximum/minimum value if a subject made the same/opposite choice as the ideal agent's in all trials, regardless of complexity condition changes.

Full details of choice optimality are provided in Supplementary Methods—Behavioral measure (choice optimality).

**Computational model of arbitration.** Computational models of arbitration used in this study are based on the previous proposal of arbitration control[30]. The original arbitration model uses a dynamic two-state transition[31] to determine the extent to which the control is allocated to a model-based learner (MB)[24] and to a model-free SARSA learner (MF)[52] at each moment in time. Specifically, the change of the control weight $P_{MB}$ (the probability of choosing a model-based strategy) is given by the difference between two types of transition: MF→MB and MB→MF:

$$\frac{dP_{MB}}{dt} = \alpha(1 - P_{MB}) - \beta P_{MB}, \qquad (1)$$

where $\alpha$ and $\beta$ refer to the transition rate MF→MB and MB→MF, respectively.

The transition rate $\alpha$ (MF→MB) is found to be a function of reliability of the MF strategy that reflects the average amount of RPE[30]:

$$\alpha(x) = \frac{A}{1 + \exp(Bx)}, \qquad (2)$$

where $x$ refers to MF reliability and the two free parameters $A$ and $B$ refer to the maximum transition rate and the steepness, respectively. Likewise, the transition rate $\beta$ (MB→MF) is defined as a function of MB reliability that reflects the posterior estimation of the amount of SPE[30].

**Computational hypotheses on arbitration.** We tested three computation types on arbitration incorporating prediction uncertainty and task complexity.

First, to test "goal-driven MF" (Fig. 5a) we implemented the following versions: (1) "1MF model" refers to the null hypothesis that the MB and the goal-independent MF interacts; (2) "3Q model" refers to the hypothesis that the MB interacts with a single MF with goal-dependent state-action value sets. Specifically, the MF learns a state × action × goal(red/blue/silver) value matrix with a single learning rate; (3) "3MF model" refers to the hypothesis that the MB interacts with goal-dependent multiple MFs. Specifically, each goal is associated with an independent MF (red, blue, and silver) with a separate learning rate.

Second, to test the effect of the state-space complexity on arbitration control, we define a transition rate as a function of both reliability and complexity (see the right box of Fig. 5a). The following variants of the transition function were used: (1) "Sign of modulation" is intended to test whether the complexity has a positive or negative influence on the transition rate. We set $z = 1$ and 2 or $z = 2$ and 1 for a low and high complexity condition, respectively. (2) "Direction of modulation" is intended to test whether the complexity influence the both transition rates MB→MF and MF→MB ("Bidirectional"), or each single transition rate ("MB→MF" and "MF→MB"). This means that the above rules (the type of interaction and the sign of modulation) are applied to both transition rates, or to a single direction, respectively. 3) "Type of interaction" is intended to investigate the effect of the task complexity on the transition rates. For simplicity, we only show the variants of the transition rate $\alpha$ (MF→MB). The same rule can be applicable to the transition rate $\beta$. "Null" assumes that there is no complexity effect on arbitration control; refer to the Eq. (2). "Interaction1" assumes that there is a direct interaction effect (complexity ($z$) × reliability ($x$)) on arbitration control.

$$\alpha(x) = \frac{A}{1 + \exp(B(1 + cz)x)}. \qquad (3)$$

"Interaction2" assumes that there is an indirect interaction effect on arbitration control. Although there is no interaction term ($zx$), the transition rate is a function of both complexity ($z$) and reliability ($x$).

$$\alpha(x) = \frac{Az}{1 + \exp(Bx)}. \qquad (4)$$

To test the effect of the sotate-space complexity on exploration, we define an

exploration parameter as a function of complexity (see the bottom-left box of Fig. 5a).

$$P(a|s) = \frac{\exp(\tau(z)\,Q(s,a))}{\sum_b \exp(\tau(z)\,Q(s,b))}. \tag{5}$$

Third, we also test the effect of complexity on exploration. To test the hypothesis that increasing complexity increases the degree of explorative choices (Fig. 5a), we set $\tau(z) = 1$ and 0.5 for the low and high complexity condition, respectively. For testing the hypothesis that increasing complexity decreases the degree of explorative choices, we set $\tau(z) = 0.5$ and 1.

Note that we compared prediction performance of all combinations of the above cases, and for simplicity we showed the results of only 41 major cases (Fig. 5b); in most of cases prediction performance is far below than the stringent threshold (exceedance probability $p = 1e-3$).

**Comparison between our model and a linear MB-MF mixture.** In a stable environment (i.e., a fixed amount of state-transition uncertainty and a fixed level of task complexity), the state of our computational model converges to a fixed point. This is specified by the steady-state model choice probability:

$$\frac{dP_{MB}}{dt} = 0. \tag{6}$$

Then by using (1), we get

$$P_{MB} = \frac{\alpha}{\alpha + \beta}, \; P_{MF} = \frac{\beta}{\alpha + \beta}. \tag{7}$$

Note that this is the equivalent of a simple mixture of MB and MF RL[7].

**fMRI data acquisition.** Functional imaging was performed on a 3T Siemens (Erlangen, Germany) Tim Trio scanner located at the Caltech Brain Imaging Center (Pasadena, CA) with a 32 channel radio frequency coil for all the MR scanning sessions. To reduce the possibility of head movement related artifact, participants' heads were securely positioned with foam position pillows. High-resolution structural images were collected using a standard MPRAGE pulse sequence, providing full brain coverage at a resolution of 1 mm × 1 mm × 1 mm. Functional images were collected at an angle of 30° from the anterior commissure–posterior commissure (AC–PC) axis, which reduced signal dropout in the orbitofrontal cortex. Forty-five slices were acquired at a resolution of 3 mm × 3 mm × 3 mm, providing whole-brain coverage. A one-shot echo-planar imaging pulse sequence was used (TR = 2800 ms, TE = 30 ms, FOV = 100 mm, flip angle = 80°).

**fMRI data analysis.** The SPM12 software package was used to analyze the fMRI data (Wellcome Department of Imaging Neuroscience, Institute of Neurology, London, UK). The first four volumes of images were discarded to avoid T1 equilibrium effects. Slice-timing correction was applied to the functional images to adjust for the fact that different slices within each image were acquired at slightly different points in time. Images were corrected for participant motion, spatially transformed to match a standard echo-planar imaging template brain, and smoothed using a 3D Gaussian kernel (6 mm FWHM) to account for anatomical differences between participants. This set of data was then analyzed statistically. A high-pass filter with a cutoff at 129 s was used.

**GLM design.** A GLM was used to generate voxel-wise statistical parametric maps from the fMRI data. We created subject-specific design matrices containing the following regressors: (R1) regressors encoding the average BOLD response at two choice states and one outcome states, (R2, R3) two parametric regressors encoding the model-derived prediction error signals—SPE of MB and RPE of MF, (R4) a regressor encoding the average BOLD response at the start of each choice state (the time of presentation of the values of each token in the first stage and the time of the state presentation in the second stage), (R5) a parametric regressor encoding the goal change; it is a binary variable indicating whether the type of a coin associated with the largest value is different from the one in the previous trial. (R6) a parametric regressor encoding max or separate reliability of MB and MF, (R7) a parametric regressor encoding complexity, (R8) a parametric regressor encoding complexity × max reliability, (R8, R9) two parametric regressors encoding the chosen value of the model-free and the model-based system, respectively ($Q_{MF}$ and $Q_{MB}$), (R10) and one parametric regressor encoding the chosen minus the unchosen value, a weighted sum of the $Q_{MB}$ and $Q_{MF}$ values according to the output of the arbitration system ($Q_{Arb}$). For value signals of the arbitration output, we also in a separate model tested for the effects of both the chosen values alone instead of the effect of chosen minus unchosen value, but as found previously in our 2014 paper, we found that the chosen minus unchosen value signal showed a more robust effect in vmPFC, hence we used chosen vs unchosen value for the arbitration value signal in our main fMRI analysis. For each GLM run at the single subject level, orthogonalization of the regressors was disabled. Finally, we implemented a standard second-level random effects analysis for each regressor of interest, and applied correction for multiple comparisons. Specifically, after running the first level GLM including all the regressors of interest, we ran a one-sample *t*-test at the second level for each separate regressor (i.e., random effects model each in a separate). Our primary means of correction was small volume correction

using 10 mm spheres centered on the coordinates for the relevant computational signals from our 2014 study[30], given we had strong a priori hypotheses about the location of each of the computational variables based on our original study. However, we also reported if the clusters survived more stringent correction at the whole brain level, cluster corrected at $p < 0.05$ FWE (extent threshold at $p < 0.001$), or the more stringent again whole brain voxel-level correction at $p < 0.05$ FWE.

**Bayesian model selection analyses on fMRI data.** To formally test which version of arbitration control provides the best account of responses in ilPFC, we ran a Bayesian model selection[33]. We chose three models—$\{\alpha,\beta\}$, $\{\alpha,\beta\}_2$, $\{\alpha_2,\beta\}$, the original arbitration model[30] and the two other versions that we found in Bayesian model selection analysis on behavioral data exhibit the second best and the best performance, respectively. We used a spherical ROI centered on the coordinates $(-54, 38, 3)$ and $(48, 35, -2)$ from the previous study[30] with a radius of 10 mm.

**Reporting summary.** Further information on research design is available in the Nature Research Reporting Summary linked to this article.

## Data availability

The raw behavioral data and fMRI results are available for download at https://github.com/brain-machine-intelligence/task_complexity_2018.

## Code availability

The simulation codes are also available for download at https://github.com/brain-machine-intelligence/task_complexity_2018.

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

## Acknowledgements

We thank Peter Dayan for insightful comments and Ralph Lee for his assistance. This work was funded by grants R01DA033077 and R01DA040011 to J.P.O.D. from the National Institute on Drug Abuse. This work was also supported by Institute of Information & Communications Technology Planning & Evaluation(IITP) grant funded by the Korea government (MSIT) (No. 2019-0-01371, Development of brain-inspired AI with human-like intelligence) (No. 2017-0-00451, Development of BCI based Brain and Cognitive Computing Technology for Recognizing User's Intentions using Deep Learning), the ICT R&D program of MSIP/IITP (No. 2016-0-00563, Research on Adaptive Machine Learning Technology Development for Intelligent Autonomous Digital Companion), the National Research Foundation of Korea (NRF) grant funded by the Korea government (MSIT) (No. NRF-2019M3E5D2A01066267), and Samsung Research Funding Center of Samsung Electronics under Project Number SRFC-TC1603-06.

## Author contributions

S.W.L. and J.P.O.D. conceived and designed the study. S.W.L. implemented the behavioral task and ran the fMRI study. D.K., G.Y.P., and S.W.L. designed computational models and analyzed the data. S.W.L., J.P.O.D., and D.K. wrote the paper. All authors approved the final version for submission.

## Competing interests

The authors declare no competing interests.
