## [Peer Review File · Nature Communications]

Reviewers' comments:

Reviewer #1 (Remarks to the Author):

In this study, Kim and colleagues devised a two-stage Markov decision task in which state-transition uncertainty (low vs high) and task complexity (low vs high) were systematically varied. The logic behind the design was state prediction errors would be abundant under high uncertainty, promoting a model-free strategy; while model-based would be preferential under circumstances of low uncertainty. Model comparison analysis of >100 possible models revealed a single model that captured behavioral performance. In this model, increasing complexity increased the likelihood of transitioning to a model-based strategy, but that increased complexity coupled with increased uncertainty favored a model-free strategy. While performing the Markov decision task, 22 subjects were scanned with fMRI. fMRI analyses replicated a number of previous findings. The novel finding was that, when compared to a prior study only examining uncertainty, signals from the model incorporating reliability and uncertainty prevailed over that only taking uncertainty into account. While no a priori regions showed a main effect of complexity, complexity and reliability interacted in the inferior lateral prefrontal cortex.

The Markov task design is clever and the Bayesian model selection analysis provides a rigorous approach to model testing. That said, there are two main weaknesses. No descriptive account of participant behavior is provided. This omission is particularly surprising given that the experimental design is one of the more innovative components of the study. Perhaps the biggest issue is the incremental nature of the finding. The experiment heavily references a previous study (Lee et al. 2014) which manipulated only uncertainty and examined model-free/model-based arbitration.

This manuscript read as if the goal was to see how the inclusion of complexity changed or better accounted for the patterns observed in Lee et al 2014. Specific comments below.

Main points

1. The rationale for systematically varying complexity was to provide a simple means of varying computational cost. The idea was that high complexity equaled high computational cost and that such cost should promote model-free (MF) mechanisms. However, in this particular Markov decision task, increasing task complexity actually promoted a model-based (MB) mechanism. This is not a problem in and of itself. However, this potentially means that instead of investigating computational cost, as was the original intent, the authors were investigating the factor of complexity. This factor may or may not be related to computational cost. This concern would be reduced if there was a way to objectively measure computational cost or if other factors that should promote computational cost would produce the same behavioral and neural effect as increasing task complexity.

2. Key to the study's innovation is the Markov decision task and the reader is treated to a detailed account of the task (Figure 1). This was very helpful and allowed me to visualize not only the task but the state-space in which the task operates. However, no descriptive analysis of the behavior is performed and no direct measure of behavior is reported. This prevents any deeper thinking/understanding of the neural results. Later, a proxy for behavior (choice optimality) is shown but the authors even point out that 'model fits' are being shown. These model fits are not directly indicative of the participant's actual behavior. The manuscript would be better served by either removing Figure 2 and replacing it with descriptors of participant behavior or at least adding the behavior to Figure 2.

3. Large sections of the results are dedicated to showing replications of previous studies, e.g. p. 14 "Neural representations of MB and MF RL" and p. 14/15 "Arbitration signals in prefrontal cortex". The initial results of the first section are replications and it appears that some of the later results are new. But it was hard to tell. Replicating key findings is a strength but these results should be

clearly divided from the novel findings being reported in the current data set.

4. The Bayesian model selection analysis was thorough and it was impressive that it settled on a single model. However, comparison to 117 other models seems like overkill. Is it possible that by simultaneously comparing this number of models, many of which were highly similar to one another, that these would effectively compete against one another? If this is not possible, it would be helpful to include a description of why this is the case. If the model were run on only the five models with the highest exceedance probabilities would the identified model prevail to the same degree?

5. It does not seem surprising that when a task manipulates uncertainty and complexity, a model that only incorporates uncertainty (as in Lee et al 2014) more poorly accounts for iIPFC/vmPFC activity compared to a model using an uncertainty x complexity interaction. Does the current model provide for a better account of the iIPFC/vmPFC activity pattern observed in the Lee et al 2014 study? If yes, then the current model is an advance. If not, it is more likely that the current model only better captures iIPFC/vmPFC activity in this study.

For example, imagine designing a Markov decision task that systematically varied trial density (high vs low). This would likely impact MB and MF arbitration. Bayesian model testing would likely identify a single model that best captured behavior. Would that model most likely be different than of Lee et al 2014 and the current study? If so, what would this tell us about the nature of arbitration in prefrontal cortex? I don't bring this up to be nitpicky. The core aspects of this experiment are solid (fMRI analysis, Markov design, model testing). I am just wondering how to best use this approach to advance understanding of prefrontal arbitration rather than producing a model tailored to the specific finding of an experiment.

Minor points

1. F statistics and degrees of freedom are never provided for ANOVA results.

2. The main effect of uncertainty and the complexity x uncertainty interaction for the model preference (Figure 4A) are not visually apparent visually. Rerunning the ANOVA and report F statistic and p values would be prudent.

Reviewer #2 (Remarks to the Author):

This manuscript presents data on a very timely and exciting topic, namely the arbitration process between model-free and model-based RL. This is an extension of previous work by the authors (Lee et al., *Neuron*, 2014). In general, the experimental set out including manipulations of state-uncertainty and task complexity (computational cost via choice availability) is plausible and a logical next step. The neural data presented could potentially be highly informative. My major concern with this paper, which substantially limits my initial enthusiasm, is one that (on the second glance) disappointed me with its ancestor: there is lack of a clear link between the model predictions and distinct behavioural readouts. This being combined with what the authors refer to as 'large scale' model comparison is somehow more worrisome. While the use of such model selection techniques is overall supported, the necessity to a priori demonstrate the distinct prediction of models or classes of models related to distinct features in the data still remains inevitable. In short, the manuscript as presented (and maybe the task in general) misses a clear presentation of distinct model-free and model-based behavioural readouts. It could be that I missed some aspects of distinct predictions possible on this task because of the way the authors present their data (by neglecting my main critique). I will elaborate on this issue in detail below. The fMRI data presented could be very interesting, however, the issues regarding their overall

approach to distinct behavioural predictions based on the model(s) needs to be resolved first as their fMRI analyses is essentially based on model predictions inferred from the choice data.

Behavioural readout:

As noted, there is a lack of a clear and distinct behavioural readout per condition (at least in the way the data is presented).

- First, and surprisingly, choosing right or left on the first actions somehow does not matter (at least from the model-based perspective) because all possible outcomes can be reached from each second state. The second state is designed to show a sensitivity to the (on every trial) instructed value. Thus, based on well-known effects of reinforcement on choice repetition through model-free control, I assume there will be an effect of reward in the previous trial on the first action (e.g. repeating R after choosing a R and L1 sequence that got rewarded), which would thus constitute a measure of model-free control as a model-based controller knows that it could have chosen L as first action to reach the same outcome with same likelihood.

- Second, depending on having only some outcomes available at a second state and still their value being instructed at the beginning of each trial, a model-based controller could make a specific choice for the first action by planning ahead to reach a certain outcome, thus, resulting in a specific measure of model-based control (I am not sure whether this is possible at all based on their design: so far, I believe not)

- Having established clear behavioural readouts of each controller (e.g. in logistic regression model on choice repetition), one can test straightforward the influence of uncertainty and complexity on these.

- Subsequently, making very clear, which parameter(s) in their model(s) influence(s) what kind of behaviour (and how) seems necessary and it should be straightforward to be demonstrated in simulations.

- The "high uncertainty condition" with 50/50 transitions remains conceptually unclear to me. While it is obvious that this induces high levels of uncertainty and reduces model-based control, the authors mention themselves that this results in a random transition, thus, rendering any kind of model-based control meaningless because there is essentially no structure to be detected. I can see why this is under certain circumstances an interesting condition to be included but think that for demonstrating their argument using an additional shift from 90/10 to 70/30 or 60/40 would have been more informative because model-based behaviour (if distinctly detectable with this task at all) would still have been possible

- The way how the exact number (or range) of trials per block and per condition was determined cannot be clearly followed from the manuscript. Please specify in a way that other researcher could reproduce the task based on the manuscript

If the authors can show clear distinct behavioural readouts for model-free and model-based control in this task (or could present a stringent and thus convincing argument why they feel this is not necessary), they still need to demonstrate whether their inferred parameters can actually recover the key behavioural features of their task. This is essentially necessary. Please also include the per-subject negative log-likelihood to the supplemental table showing model parameters and please use histograms or a table with percentiles to show distribution of parameters and the likelihood (and there measure of model evidence). Although it is very much appreciated that the authors share behavioural data and code, there is no word on how they infer parameters from the behavioural data and how they approximate or estimate the log model evidence, which they have to enter in the RFX BMS.

As mentioned before, the fMRI results could be very interesting but I refrain to comment on them because all fMRI analyses crucially rest upon regressors extracted from the model.

I hope the authors can find these comments helpful as they are not meant to devalue their work. However, I feel it is necessary to present clear behavioural readouts when talking about model-free and model-based control and their arbitration. I believe the authors might agree on this perspective. Distinct behavioural readouts have already been challenging to understand in their previous paper from 2014.

Reviewer #3 (Remarks to the Author):

Kim et al. present a study that builds on previous work from Lee and O'Doherty investigating the arbitration between putative model-free and model-based controllers. In this study they additionally modulate the "complexity" of the task, by manipulating the number of choice options (low: 2 vs high: 4) at the second choice stage of the task. This leads to a 2x2 factorial design in which state uncertainty (low: 0.5/0.5 vs high: 0.9/0.1) is crossed with complexity. Behaviorally they show that somewhat counterintuitively, increased complexity (defined in this way) leads to more model-based control, whereas increased uncertainty leads to more model-free control, as expected. Furthermore, they show that there is a logical interaction which shows that with greater uncertainty the effect of complexity is attenuated, unsurprisingly. This is an interesting pattern of behavioral patterns that adds to the field's understanding of these putative control systems. In their fMRI results, they first nicely replicate their previous work showing reliability effects in bilateral iIPFC and integrated value signals (which in effect reflect a comparison between model-based and model-free controllers) in vmPFC. They then show that a) their new model that incorporates task complexity better accounts for the BOLD signals in these areas; and b) there is an interaction with task complexity, although I do have a query about its interpretation. Overall, this is an impressive, rigorous study that advances our understanding of the conditions under which model-based versus model-free systems win out to control behavior, a topic likely to be of interest to a broad audience.

More details of the task could be provided and I have a few questions about the task/model:

How often does the task transition between high and low states of uncertainty? Do the interaction effects depend on whether this has just happened or many trials earlier (in particular for the transition from high to low uncertainty)?

How is the goal change regressor defined?

I could not work it out but does not task complexity influence the computation of reliability in the model, since higher number of choice options should increase the entropy of the choice? Is this relation already taken into consideration in the computation of reliability in the model?

Figure 7 suggests that the iIPFC effect goes in the opposite direction of the behavioral results. When complexity is higher, the MB reliability signal is lower, and vice versa? How should we conceptualize this interaction then?

Behaviorally the results show that somewhat counterintuitively, increased complexity leads to more model-based control. This suggests that having to plan more carefully, or to consider more options, actually makes people more model-based. A suggestion I hope will be helpful: it would be interesting to explore at what point this breaks down due to over-taxed cognitive demands.

Editor's comment (on the issue raised by the reviewer#1 and #2)

We feel it will be important to address the question of the potential mismatch between model prediction and behavioral readout as raised by Reviewers #1 and #2. We will need this and other technical concerns of the referees to be alleviated by the next round of re-review.

>> We appreciate both the editor and the reviewers' insightful comments about behavioral readout. To fully address this concern, we have provided a clear read-out for model-based control, called choice optimality. As a result, we have added three new figures (Figure 2, Figure 5, and one supplementary figure; see below).

1. Behavioral readout of model-based control - subjects' data (Figure 2)

We showed in an independent computer simulation that the choice optimality measure allows us to not only distinguish between model-based and model-free control (Fig. 2A; shown below), but also successfully separate out the effect of model-based control from model-free for each experimental condition (Fig. 2B).

In a behavioral analysis in which we computed choice optimality of subjects' actual behavior data, we found that the behavioral profile of subjects (Fig. 2C) are highly consistent with our prediction (Fig. 2B). We also found that the two experimental variables, uncertainty and complexity, are the two key factors that influence choice optimality (Fig. 2D).

These results fully support our hypothesis that subjects combine model-based and model-free control during performing the task, and the degree of engagement of model-based control is affected by both uncertainty and complexity.

Behavioral readout of model-based control (computer simulation)

Behavioral readout of model-based control (subjects' data)

Figure 2. Behavioral results. (A) Choice optimality (a proxy for assessing the degree of agents' engagement in model-based control), of a model-based and model-free reinforcement learning agent. Choice optimality depicts the degree of match between agents' actual choices and an ideal agent's choice corrected for the number of available options. For full details of this measure, refer to Experimental Procedures. **(B)** Difference in choice optimality between a model-based and model-free reinforcement learning agent for the four experimental conditions (low/high state-transition uncertainty x low/high task complexity). Shown in red boxes are the effect of the two experimental variables on each measure (2-way repeated measures ANOVA). **(C)** Participants' choice optimality for the four experimental conditions. Shown in red boxes are the effect of the two experimental variables on each measure (2-way repeated measures ANOVA; also see Table S3 for full details). **(D)** Results of a general linear model analysis (dependent variable: choice optimality, independent variables: uncertainty, complexity, reward values, choices in the previous trial, and goal values). Uncertainty and complexity, the two key experimental variables in our task, significantly influence choice optimality (t -test; $p < 0.001$). Error bars are SEM across subjects.

2. Behavioral readout of model-based control - model's prediction (Figure 5)

Next, we found that the subjects' choice optimality patterns (Fig. 2C) are predicted by our computational model (Fig. 5A). In addition to the model's behavioral prediction, we also examined the latent variable of the computational model that represents control weights allocated to the model-based system, and showed that this behavioral pattern might be

attributable to engagement of model-based control in task performance affected by uncertainty and complexity (Fig. 5B).

We further checked whether the model's free parameters encapsulate the effect of uncertainty and complexity on choice optimality. For this, we ran a parameter recovery analysis. The parameter recovery analysis evaluates the degree of consistency between data-to-model parameter and model parameter-to-data conversion. It consists of the following sequence of processes: Subjects' data → model fitting 1 (original parameters) → simulated data → model fitting 2 (recovered parameters). Note that the two model fittings are independent processes; the model fitting 1 is performed on the participants' behavioral data while the model fitting 2 is performed on the behavior of the model obtained from the model fitting 1. The simulated data was generated by running simulations with the best fitting model performing on the behavioral task.

If the model parameter(s) successfully encode the key features of behavior, then the corresponding parameters should be well recovered. In other words, the original and the recovered parameter values should be significantly correlated. If the model's predictions are based on less important dimensions of behavior (e.g., overfitting), then the principal dimensions of the simulated data would likely diverge from those of the original subjects' data, leading to an unsuccessful recovery of parameter values.

We found that the four key parameters of the model (the sensitivity of reward and state prediction error, learning rates for model-based and model-free control, exploration sensitivity) are necessary for accounting for subjects' behavior patterns (Fig. 5C and 5D). The significant correlations between the original and the recovered parameter values highlight the necessity of the key parameters of the model, as well as helping to rule out a substantive effect of overfitting. Moreover, our computational model exhibits the best explanatory account of this effect, compared to alternative models (Supplementary Figure S6). This fully establishes a link between model parameters and choice optimality.

Behavioral readout of model-based control (computational model)

Effect of uncertainty and complexity on model's choice optimality (Behavior recovery analysis)

Figure 5. Computational model fitting results. (A) Patterns of choice optimality generated by the best fitting version of the arbitration model, using parameters obtained from fitting to participants behavior. For this, the model was run on the task (1000 times), and we computed choice optimality measures in the same way as in Fig. 2B and 2C. (B) Degree of engagement of model-based control predicted by the computational model, based on the model fits to individual participants. P_{MB} corresponds to the weights allocated to the MB strategy. Shown in the red box are the effect of the two experimental variables on each measure (2-way repeated measures ANOVA; also see Table S4 for full details). Error bars are SEM across subjects. (C, D) Behavioral effect recovery analysis. The individual effect sizes of uncertainty (C) and complexity (D) on choice optimality of subjects (true data) were compared with those of our computational model (simulated data).

A Effect of uncertainty and complexity on model's choice optimality (Behavior recovery analysis)

B Uncertainty effect (comparison between behavioral and simulation data)

C Complexity effect (comparison between behavioral and simulation data)

Figure S6. Related to Figure 5. (A) The figures show the effect size of uncertainty and complexity on choice optimality of different models, including the best version of the model incorporating both uncertainty and complexity (Our model), the model incorporating uncertainty only (Lee2014), a pure model-based agent (Model-based), a pure model-free agent (Model-free). The effect sizes were computed by running a general linear model analysis with the choice optimality being included as a dependent variable, and uncertainty, complexity, reward values, choices in the previous trial, and goal values as independent variables (the same way as in Figure 5). The uncertainty and complexity, the two experimental variables of our task, are the two key factors that influences choice optimality (t -test; $p < 0.001$). Error bars are SEM across subjects. **(B)** Behavioral effect recovery analysis. The individual effect sizes of uncertainty/complexity on choice optimality of subjects (behavioral data) were compared with those of each model (simulated data).

Reviewers' comments:

Reviewer #1 (Remarks to the Author):

In this study, Kim and colleagues devised a two-stage Markov decision task in which state-transition uncertainty (low vs high) and task complexity (low vs high) were systematically varied. The logic behind the design was state predictions errors would be abundant under high uncertainty, promoting a model-free strategy; while model-based would be preferential under circumstances of low uncertainty. Model comparison analysis of >100 possible models revealed a single model that captured behavioral performance. In this model, increasing complexity increased the likelihood of transitioning to a model-based strategy, but that increased complexity coupled with increased uncertainty favored a model-free strategy. While performing the Markov decision task, 22 subjects were scanned with fMRI. fMRI analyses replicated a number of previous findings. The novel finding was that, when compared to a prior study only examining uncertainty, signals from the model incorporating reliability and uncertainty prevailed over that only taking uncertainty into account. While no a priori regions showed a main effect of complexity, complexity and reliability interacted in the inferior lateral prefrontal cortex.

The Markov task design is clever and the Bayesian model selection analysis provides a rigorous approach to model testing. That said, there are two main weaknesses. No descriptive account of participant behavior is provided. This omission is particularly surprising given that the experimental design is one of the more innovative components of the study. Perhaps the biggest issue is the incremental nature of the finding. The experiment heavily references a previous study (Lee et al. 2014) which manipulated only uncertainty and examined model-free/model-based arbitration.

This manuscript read as if the goal was to see how the inclusion of complexity changed or better accounted for the patterns observed in Lee et al 2014. Specific comments below.

Main points

1. The rationale for systematically varying complexity was to provide a simple means of varying computational cost. The idea was that high complexity equaled high computational cost and that such cost should promote model-free (MF) mechanisms. However, in this particular Markov decision task, increasing task complexity actually promoted a model-based (MB) mechanism. This is not a problem in and of itself. However, this potentially means that instead of investigating computational cost, as was the original intent, the authors were investigating the factor of complexity. This factor may or may not be related to computational cost. This concern would be reduced if there was a way to objectively measure computational cost or if other factors that should promote computational cost would produce the same behavioral and neural effect as increasing task complexity.

>> Reply 1-(1).

We appreciate the reviewer's thoughtful comments about computational cost. The original intention of this study was to study computational cost, but we acknowledge that we focus on complexity and uncertainty as one potential way to manipulate computational cost and that our

findings may not be deemed to generalize to other forms of computational cost. To reflect the reviewer's concern, we have added caveats and toned down claims of the paragraph in the discussion session as follows:

"We also acknowledge that we focus on complexity and uncertainty as one potential way to manipulate computational cost and that our findings may not be deemed to generalize to other forms of computational cost. Future studies could therefore focus on more clearly separating these effects of computational load from other factors that may be related to computational cost."

To further resolve the reviewer's concern, we have tried to quantify the effect of complexity x uncertainty on the computational cost. There is no unique way of numerically coding or quantifying computational cost, so our task design introduced three variables as a proxy for this: the size of the state space, action space, and goal space, each of which can be numerically coded as the number of nodes, routes, and available coins, respectively.

Shown below are the four experimental conditions (uncertainty x complexity) and corresponding quantitative assessments of computational cost (see the green rounded box). Although the quantification may depend on the agent's ability (e.g., agent's sensitivity to state-transition probability), it clearly shows that our task is designed to simulate conditions with different levels of computational load (the total computational load varying between 15 and 34). To the best of our knowledge, this is the first attempt to manipulate computational cost by using a combination of an explicit (complexity) and an implicit (uncertainty) task variable. To reflect this, we have updated the Figure S1 (see below) and have added details to its caption.

Figure S1. Illustration of four different types of conditions in the task (low/high x uncertainty/complexity) Shown in the rounded green box are the quantitative assessment of computational cost in each condition. Since there is no unique way of numerically coding or quantifying computational cost, our task design introduced three variables as a proxy for this: the size of state space, action space, and goal space, each of which can be numerically coded as the number of nodes, routes, and available coins, respectively. The estimated computational costs show that our task design can simulate conditions with different levels of computational load (the total computational load varying between 15 and 34).

2. Key to the study's innovation is the Markov decision task and the reader is treated to a detailed account of the task (Figure 1). This was very helpful and allowed me to visualize not only the task but the state-space in which the task operates. However, no descriptive analysis of the behavior is performed and no direct measure of behavior is reported. This prevents any deeper thinking/understanding of the neural results. Later, a proxy for behavior (choice optimality) is shown but the authors even point out that 'model fits' are being shown. These model fits are not directly indicative of the participant's actual behavior. The manuscript would be better served by either removing Figure 2 and replacing it with descriptors of participant behavior or at least adding the behavior to Figure 2.

>> Reply 1-(2).

Thank you very much for the constructive suggestion.

To fully reflect the reviewer's concern, we have added a new figure (Figure 2; see below) providing a descriptive analysis of behavior associated with model-based control (choice optimality). Full details of these behavioral measures are provided in Experimental Procedures (see below).

In Figure 2, we used a computer simulation with pure model-based and model-free reinforcement learning agents to demonstrate that choice optimality is a good proxy of the extent to which participants engage in model-based control (Figure 2A), and that it is sensitive to the two experimental variables of our task: uncertainty and complexity (Figure 2B). This is confirmed by the choice optimality of participants' behavior (Figure 2C). Moreover, both uncertainty and complexity are the two main factors that influences choice optimality (Figure 2D).

Behavioral readout of model-based control (computer simulation)

Behavioral readout of model-based control (subjects' data)

Figure 2. Behavioral results. (A) Choice optimality (a proxy for assessing the degree of agents' engagement in model-based control), of a model-based and model-free reinforcement learning agent. Choice optimality depicts the degree of match between agents' actual choices and an ideal agent's choice corrected for the number of available options. For full details of this measure, refer to Experimental Procedures. (B) Difference in choice optimality between a model-based and model-free reinforcement learning agent for the four experimental conditions (low/high state-transition uncertainty x low/high task complexity). Shown in red boxes are the effect of the two experimental variables on each measure (2-way repeated measures ANOVA). (C) Participants' choice optimality for the four experimental conditions. Shown in red boxes are the effect of the two experimental variables on each measure (2-way repeated measures ANOVA; also see Table S3 for full details). (D) Results of a general linear model analysis (dependent variable: choice optimality, independent variables: uncertainty, complexity, reward values, choices in the previous trial, and goal values). Uncertainty and complexity, the two key experimental variables in our task, significantly influence choice optimality (t-test; $p < 0.001$). Error bars are SEM across subjects.

“Behavioral measure (choice optimality). Choice consistency, a conventional behavioral measure used to quantify insensitivity to changes in the environmental structure (one of the key characteristics of model-free RL), works well for conventional two-step task paradigms in which the environment is stable for a certain period [Daw, Neuron 2011; Miller, Nature neurosci 2017]. Unfortunately, it is not a suitable measure for our highly dynamic task design in which we manipulate task complexity for the following reasons: First, reward values fluctuate on a trial-by-trial basis. This manipulation encourages trial-by-trial arbitration between model-free and model-based control. Choice behavior on

each trial is affected by the relative values of each coins, nullifying the choice consistency effect. Second, the level of state-space complexity also varies on a trial-by-trial basis. State-space complexity is manipulated by varying the number of available choices. The choice consistency rate would plummet when the number of available choices increases from 2 to 4. Third, the independent manipulation of the first two factors (state-space complexity and reward value) further promotes arbitration. For example, if the values of the three coins remain constant on each trial, then it's likely that choice behavior would converge to a specific sequence of choices for each task complexity condition, which does not necessitate arbitration control. Fourth, most of the goals are achievable in state1 regardless of experimental conditions. This means that there are usually more than two different behavioral policies or pathways/outcome states that enable a subject to achieve a goal (coin). These factors make it difficult to apply choice consistency to the present task design.

To deal with all the above issues, we devised an alternative behavioral measure that is robust against the above-mentioned experimental issues: choice optimality. This measure quantifies the extent to which participants on a given trial took the objectively best choice had they complete access to the task state-space, and a perfect ability to plan actions in that state-space. It is based on the choice of the ideal agent assumed to have a full, immediate access to information of the environmental structure, including state-transition uncertainty and task complexity. The choice optimality is defined as the degree of match between subjects' actual choices and an ideal agent's choice corrected for the number of available options. To compute the degree of choice match between the subject and the ideal agent, for each condition, we calculated an average of normalized values (i.e., likelihood) of the ideal agent for the choice that a subject actually made on each trial. To correct for the number of options, we then multiplied it by 2 for the high complexity condition; this is intended to compensate for the effect that the baseline level of the likelihood in the high complexity condition (# of available options =4) becomes half of that in the low complexity condition (# of available options =2). In other words, this adjustment effectively compensates the effect of # of available options on normalization without biasing the correspondence between participant's choices and optimal choices. The choice optimality value would have a maximum/minimum value if a subject made the same/opposite choice as the ideal agent's in all trials, regardless of complexity condition changes.

Owing to the fact that the ideal agent's behavioral policy is not affected by the variability of such experimental variables, this measure serves as a reasonable proxy for assessing the degree of participants' engagement in model-based control. In principle, provided that the model-based agent has complete knowledge of the state-space and no cognitive constraints, it will always choose more optimally than a model-free agent."

Please note that this figure is followed by a computational hypothesis (Figure3), model comparison (Figure 4), and model-based analyses (Figure 5) in which we fully establish a link between all these behavioral measures and the model (refer to our reply 2-(3)).

In addition to our analysis on choice optimality, we have added another supplementary figure showing conventional behavioral measures for reward-based learning (Figure S2). Note that this measure also allows us to distinguish model-based control from model-free control (Figure S2A), and we found evidence in subjects data to fully dismiss the possibility that subjects use a pure model-free control strategy. The results from that additional analysis are also fully consistent with the predictions of our model, suggesting that our computational model encapsulated the essence of subjects' choice behavior (Figure S2B; see below). Full details of these measures are provided in Supplementary Methods.

Figure S2. (A) Predicted choice bias patterns of model-free (MF) and model-based (MB) control, calculated for the three goal conditions defined as the trials according to which coin has the maximum monetary outcome value (low, medium, and high token value for the L choice). Owing to the asymmetric association between outcome states and coin types (For full details, see Supplementary Methods – Choice bias), participants would exhibit distinct choice bias patterns for each goal condition that distinguishes model-based from model-free control; the MF control agent would exhibit a balanced choice bias pattern, whereas the MB control agent would show a slight left bias pattern. For full details of this measure, refer to Supplementary Methods - Behavioral measure. **(B)** Participants' choice bias and choice consistency, the conventional behavioral markers indicating reward-based learning. Error bars are SEM across subjects. The prediction about the choice bias matches subjects' actual choice bias (the left of the below figure). In particular, the data shows a clear left bias pattern, rejecting the null hypothesis that subjects used a pure model-free control strategy. This bias is also reflected in choice consistency (the right plot). These results also indicate that participants' choice behavior is guided by reward-based learning more generally. **(C)** Choice bias (left), choice consistency (middle), and the average value difference (right) of our computational model of arbitration control (Figure 4B). For this, we ran a deterministic simulation in which our computational model experiences exactly the same episode of events as each individual subject, and we generated the trial-by-trial outputs. The max goal conditions are defined in the same way as in the Figure 2A. Error bars are SEM across subjects. Note that both the choice bias and choice

consistency patterns of the model (the left and the middle plot) are fully consistent with the behavioral results (B). Second, the values difference (right – left choice) of the model is also consistent with this finding (the right plot), suggesting that these behavioral patterns are originated from value learning. In summary, our computational model encapsulates the essence of subjects' choice behavior as guided by reward-based learning.

“Behavioral measure (choice bias). In our task design, a left/right choice bias in the first stage can be interpreted as a behavioral marker indicating reward-based learning.

Our task design involves delicate manipulation of the goal. Note that the association between goal types and coin colors were randomized for each subject, and here we show one particular example (see the above figure). Let's define the R branch as the bottom left routes (accessible by making the R choice in the first stage) and the L branch as the top right routes (accessible by making the L choice in the first stage), respectively. The agent is not informed about task complexity until the second stage, so in the first stage a rational agent would make the following assumptions: in the first stage, outcome states associated with a silver coin are accessible by making a primary choice in both branches. An outcome state associated with a red coin are accessible by making a primary and a secondary choice in the R and L branch, respectively. An outcome state associated with a blue coin are accessible by making a secondary choice in both the R and the L branch.

Accommodating this situation, we can roughly calculate the expected value of the L/R choice of the optimal agent (the same agent used to compute choice optimality) for the first stage. The probability of transitioning to a desired outcome state by making a primary and a secondary choice is given by (0.7, 0.3), which is computed by taking average of the two state-transition probability values: (0.9, 0.1) and (0.5, 0.5). Note that this setting is used to simulate average experimental conditions. For the sake of simplicity, reward values were normalized to 1 (for the goal coin), 0.5 (for the other coins), and 0 (unrewarded).

If we assume that an agent relies on model-free control and that the agent makes a greedy choice, we can compute the expected values and the corresponding choice biases by using the uniform state-transition probability distribution (0.5,0.5) (meaning that the agent is agnostic about state-transition uncertainty and thus cannot afford to accommodate state-transition probability value changes) as follows. Note that the low, medium, and high value coin corresponds to silver, red, and blue coins in Figure 1 and Figure S1; again, the coin colors are randomized for each subject.

- Silver coin : (the expected value of L branch) $1 \times 0.5 + 0 \times 0.5 = 0.5$. (the expected value of R branch) $1 \times 0.5 + 0.5 \times 0.5 = 0.75$. The expected value difference (L-R) = -0.25. We expect a R choice bias.

- Red coin : (the expected value of L branch) $1 \times 0.5 + 0.5 \times 0.5 = 0.75$. (the expected value of R branch) $0.5 \times 0.5 + 1 \times 0.5 = 0.75$. Therefore expected value difference (L-R) = 0. We expect no L choice bias.

- Blue coin : (the expected value of L branch) $0.5 \times 0.5 + 1 \times 0.5 = 0.75$. (the expected value of R branch) $0 \times 0.5 + 1 \times 0.5 = 0.5$. Therefore expected value difference (L-R) = +0.25. We expect L choice bias.

Therefore, if subjects performed the task using pure model-free control, they would show a well-balanced choice bias pattern: R bias, zero bias, and L bias for each goal, respectively.

On the other hand, if we assume that an agent relies on model-based control, we can compute the expected values and the corresponding choice biases, this time by using the average state-transition probability set (0.7,0.3) (meaning that the agent actively accommodates state-transition probability value changes between (0.9,0.1) and (0.5,0.5)) as follows.

- Silver coin : (the expected value of L branch) $1 \times 0.7 + 0 \times 0.3 = 0.7$. (the expected value of R branch) $1 \times 0.7 + 0.5 \times 0.3 = 0.85$. The expected value difference (L-R) = -0.15. We expect a weak R choice bias.

- Red coin : (the expected value of L branch) $1 \times 0.7 + 0.5 \times 0.3 = 0.85$. (the expected value of R branch) $0.5 \times 0.7 + 1 \times 0.3 = 0.65$. Therefore expected value difference (L-R) = +0.2. We expect a weak L choice bias.

- Blue coin : (the expected value of L branch) $0.5 \times 0.7 + 1 \times 0.3 = 0.65$. (the expected value of R branch) $0 \times 0.7 + 1 \times 0.3 = 0.3$. Therefore expected value difference (L-R) = +0.35. We expect L choice bias.

Therefore, if subjects performed the task using model-based control, they would exhibit a slight left bias pattern: weak R bias, weak L bias, and L bias for each goal, respectively.”

3. Large sections of the results are dedicated to showing replications of previous studies, e.g. p. 14 “Neural representations of MB and MF RL” and p. 14/15 “Arbitration signals in prefrontal cortex”. The initial results of the first section are replications and it appears that some of the later results are new. But it was hard to tell. Replicating key findings is a strength but these results should be clearly divided from the novel findings being reported in the current data set.

>> Reply 1-(3).

Here we clarify whether the neural results of each section are replications of the existing findings or new findings:

- Section “Neural representations of model-based and model-free RL” : replications + new findings
- Section “Arbitration signals in prefrontal cortex” : replications
- Section “Model comparison against fMRI data” : new findings
- Section “Modulation of inferior prefrontal reliability signal by complexity” : new findings

To fully reflect the reviewer’s suggestion, we have added clarifications, by making it very clear to distinguish between the replications of existing findings and novel findings in each section. For example, please refer to the below.

- In the section, Neural representations of model-based and model-free RL,
“... In summary, we replicated existing findings about variables necessary to implement MB and MR RL, including a prediction error and value signal for each system, and an integrated value signal to guide an actual choice.”
“In addition, we found new evidence for the implementation of the goal-driven MF model (Table S2; the definition of the regressor is provided in Supplementary Methods), which ...”
- In the section, Arbitration signals in prefrontal cortex,
“... These findings are again successful replications of findings from our previous study (Lee et al., 2014).”
- In the section, Model comparison against fMRI data,
“Note that these are the new findings (beyond the original findings of our 2014 study) that support our hypothesis that the model in which complexity is taken into account provides a better account of prefrontal mediated arbitration control.”
- In the section, Modulation of inferior prefrontal reliability signal by complexity,
“This provides new evidence that these two signals relevant for driving arbitration interact with each other in ilPFC.”

4. The Bayesian model selection analysis was thorough and it was impressive that it settled on a single model. However, comparison to 117 other models seems like overkill. Is it possible that by simultaneously comparing this number of models, many of which were highly similar to one another, that these would effectively compete against one another? If this is not possible, it would be helpful to include a description of why this is the case. If the model were run on only the five models with the highest exceedance probabilities would the identified model prevail to the same degree?

>> Reply 1-(4).

The reason why the Bayesian model selection (BMS) analysis settled on a single model is because the fitness values of the best version of the model are consistently higher than all the other versions across all the subjects.

In addition, as the reviewer suggested, we have now re-run the BMS on only the five best fitting models from the original BMS analysis to check if the identified model dominates the BMS analysis to the same degree. The model we identified in our original analysis is indeed the best version of the model with a very high probability ($p=0.97$; see the below figure), fully dismissing the possibility that there is an effect of competition among similar models on BMS results.

Details of the top 5 models

Model	Goal-driven M F type	Effect of complexity on transition between MB and MFRL			Effect of complexity on exploration
		Sign of modulation	Direction	Interaction	
1st	3Q	+	MF→MB	Interaction 2	Positive
2nd	3MF	+	MF→MB	Interaction 2	Positive
3rd	3MF	+	MF→MB	Interaction 2	Null
4th	1MF	+	MF→MB	Interaction 2	Null
5th	3MF	-	Bilateral	Interaction 2	Null

This result has been included in Supplementary information (Figure S4).

5. It does not seem surprising that when a task manipulates uncertainty and complexity, a model that only incorporates uncertainty (as in Lee et al 2014) more poorly accounts for iIPFC/vmPFC activity compared to a model using an uncertainty x complexity interaction. Does the current model provide for a better account of the iIPFC/vmPFC activity pattern observed in the Lee et al 2014 study? If yes, then the current model is an advance. If not, it is more likely that the current model only better captures iIPFC/vmPFC activity in this study.

>> Reply 1-(5).

We apologize for the confusion about the relationship between the models and the two tasks. This model is designed in a way that it becomes equivalent to the Lee2014 model when the complexity stays at a low constant level, which is the situation considered in the Lee 2014 study. It is apparent that the current model would make the same predictions as the original model on the 2014 dataset simply because we did not perturb complexity. So there is no need to re-run the current model in the old scenario. (A Bayesian model selection analysis would show that it performs worse on the behavioral data from 2014 because the extra degrees of freedom would be penalized in the model fitting without any advantage.) In summary,

- In the Lee 2014 task (environment with a varying degree of uncertainty), both the current model and the Lee2014 model make the same predictions.
- In the current task (environment with a varying degree of uncertainty and complexity), we showed that the current model makes significantly better predictions than the Lee 2014 model.

We designed this task to discriminate a model that is sensitive to complexity from one that is not. The main claim of our paper is that a model that incorporates complexity better accounts of vIPFC activity. We believe that this is not trivial but a real advance.

To resolve the reviewer's concern, we have added clarifications to the main text.

"Note that this model is designed in a way such that it becomes equivalent to the previous arbitration process (Lee et al., 2014) when the complexity stays at a low constant level. That is, it is apparent that the current model would make the same predictions as the original model on such a simple two-stage Markov decision task without complexity perturbation. It is also noted that when the environment is perfectly stable (i.e., a fixed amount of state-transition uncertainty and a fixed level of task complexity), the particulars of this model converge to a stable mixture of MB and MF RL (Daw et al., 2011)."

For example, imagine designing a Markov decision task that systematically varied trial density (high vs low). This would likely impact MB and MF arbitration. Bayesian model testing would likely identify a single model that best captured behavior. Would that model most likely be different than of Lee et al 2014 and the current study? If so, what would this tell us about the nature of arbitration in prefrontal cortex? I don't bring this up to be nitpicky. The core aspects of this experiment are solid (fMRI analysis, Markov design, model testing). I am just wondering how to best use this approach to advance understanding of prefrontal arbitration rather than producing a model tailored to the specific finding of an experiment.

>> Reply 1-(6).

We appreciate the reviewer's constructive comments. This is certainly an interesting point that can apply to other studies in general that rely on a combination of computational modelling and tasks designed to test variables from those models.

Please note that we started with a specific hypothesis that computational complexity (or at least task complexity) would be utilized in the arbitration process because it is relevant for the decision whether to deploy model-based or model-free inference. Given our specific hypothesis, we found at least partial support for that. Of course there are likely to be many other factors that influence arbitration — and it is going to be important to test for the effects of those other potential variables in future studies. It stands to reason that if a variable influences the arbitration process, then we would expect a model that incorporates that variables would perform better than a model that does not. This is our main claim.

Of course, if we pick an experimental manipulation that does not have any relevance to the arbitration process, then we would easily anticipate that the manipulation would fail, and the model comparison would provide no support for such a manipulation. This is of course a hypothetical argument as we haven't run that. That being said, trial density to take the reviewers' example could potentially be relevant for the arbitration because there might be "ego depletion" type effects going on for the model-based controller — in that if the agent has more time to rest between trials maybe she can utilize model-based control more effectively or something. So, if it is the case that this is relevant for the allocation of model-based vs model-free control, it is very possible we would find evidence that this is reflected in the arbitration process. This would not be a trivial result in our opinion, but an interesting one.

Minor points

1. F statistics and degrees of freedom are never provided for ANOVA results.

>> Reply 1-(7). We have provided details of statistical tests, including F statistics and degrees of freedom.

2. The main effect of uncertainty and the complexity x uncertainty interaction for the model preference (Figure 4A) are not visually apparent visually. Rerunning the ANOVA and report F statistic and p values would be prudent.

>> Reply 1-(8). We have double checked both the main effect of uncertainty and the interaction effect of complexity and uncertainty using different statistical toolboxes, and have confirmed the original ANOVA results are correct (see below). Note that Figure 4A in the previous version of the manuscript becomes Figure 5B in the revised version.

Tests of Within-Subjects Contrasts

Measure: MEASURE_1

Source	uncertainty	complexity	Type III Sum of Squares	df	Mean Square	F	Sig.
uncertainty	Linear		.008	1	.008	30.459	.000
Error(uncertainty)	Linear		.006	23	.000		
complexity		Linear	.109	1	.109	134.796	.000
Error(complexity)		Linear	.019	23	.001		
uncertainty * complexity	Linear	Linear	.003	1	.003	4.803	.039
Error (uncertainty*complexity)	Linear	Linear	.013	23	.001		

The reason it does not visually appear is that there is "high individual variability", which is fairly common in decision making tasks designed to test the effect of multiple different strategies on behavior. Note that most of individual subjects' PMB (y-axis) decreases as uncertainty increases. (see below; paired t-test $p = 1.5174e-05$).

To assure potential readers that there are significant main and interaction effects, we have made a very explicit mention of this in the main text and added an additional plot as a supplementary Table S4 including F statistics and p-values.

Reviewer #2 (Remarks to the Author):

This manuscript presents data on a very timely and exciting topic, namely the arbitration process between model-free and model-based RL. This is an extension of previous work by the authors (Lee et al., Neuron, 2014). In general, the experimental set out including manipulations of state-uncertainty and task complexity (computational cost via choice availability) is plausible and a logical next step. The neural data presented could potentially be highly informative. My major concern with this paper, which substantially limits my initial enthusiasm, is one that (on the second glance) disappointed me with its ancestor: there is lack of a clear link between the model predictions and distinct behavioural readouts. This being combined with what the authors refer to as 'large scale' model comparison is somehow more worrisome. While the use of such model selection techniques is overall supported, the necessity to a priori demonstrate the distinct prediction of models or classes of models related to distinct features in the data still remains inevitable. In short, the manuscript as presented (and maybe the task in general) misses a clear presentation of distinct model-free and model-based behavioural readouts. It could be that I missed some aspects of distinct predictions possible on this task because of the way the authors present their data (by neglecting my main critique). I will elaborate on this issue in detail below. The fMRI data presented could be very interesting, however, the issues regarding their overall approach to distinct behavioural predictions based on the model(s) needs to be resolved first as their fMRI analyses is essentially based on model predictions inferred from the choice data.

>> First of all, we greatly appreciate the reviewers' constructive comments which have helped us significantly improve our paper. We fully clarified all the issues by running behavioral, GLM, and parameter recovery analyses (please see below).

Behavioural readout:

As noted, there is a lack of a clear and distinct behavioural readout per condition (at least in the way the data is presented).

- First, and surprisingly, choosing right or left on the first actions somehow does not matter (at least from the model-based perspective) because all possible outcomes can be reached from each second state. The second state is designed to show a sensitivity to the (on every trial) instructed value. Thus, based on well-known effects of reinforcement on choice repetition through model-free control, I assume there will be an effect of reward in the previous trial on the first action (e.g. repeating R after choosing a R and L1 sequence that got rewarded), which would thus constitute a measure of model-free control as a model-based controller knows that it could have chosen L as first action to reach the same outcome with same likelihood.

>> Reply 2-(1).

We greatly appreciate the reviewer's insightful suggestion. While choice repetition is a useful behavioral measure that can in some task implementations provide a distinctive behavioral profile indicating model-free control, unfortunately, this measure is not directly applicable in our

case. In the following sections, we first explain in detail why choice repetition is not an ideal behavioral measure for our task, and then we present a more effective behavioral measure: choice optimality.

(i) Remarks on choice repetition

Since choice repetition (also known as choice consistency) is proposed to quantify insensitivity to changes in the environmental structure (one of the key characteristics of model-free RL), it would work for a conventional simple two-step task paradigm in which the environment is stable for a certain period [Daw, Neuron 2011; Miler, Nature neurosci 2017]. However, it is not directly applicable to our highly dynamic task design with a varying degree of task complexity for the following reasons:

First, reward values fluctuate on a trial-by-trial basis in the present task. This manipulation encourages trial-by-trial arbitration between model-free and model-based control. Choice behavior on each trial is affected by relative values of each coins. This feature complicates interpretation of the choice repetition effect.

Second, state-space complexity also varies on a trial-by-trial basis in our current design via manipulating the number of available choices. However, state-space complexity also complicates interpretation of a choice repetition metric, because the repetition rate necessarily decreases as the number of available choices increases from 2 to 4.

Third, in order to further promote arbitration and allow us to test for the effects of independent manipulation of the first two factors (state-space complexity and reward value) further promotes arbitration. For example, if the values of the three coins remain constant on each trial, then it's likely that choice behavior would converge to a specific sequence of choices for each task complexity condition, which does not necessitate arbitration control.

Fourth, due to uncertainty in the state-transitions and varying degree of task complexity there are usually more than two behavioral policies or pathways/outcome states that enable a subject to achieve a goal (coin). These factors made it very difficult to find a simple behavioral measure, including choice consistency or the effect of reward/state-transition type on choice repetition.

Thus in our study, the application of the choice consistency measure is confined to a simple readout of reward-based value learning guided by model-free control.

(ii) Behavioral readout : choice optimality

To deal with all the above issues, we devised an alternative behavioral measure that is robust against the above-mentioned experimental variables: choice optimality. It quantifies the extent to which participants on a given trial took objectively the best choice had they complete access to the task state-space, and a perfect ability to plan actions in that state-space. It is based on

the choice of the ideal agent assumed to have a full, immediate access to information of the environmental structure, including state-transition uncertainty and task complexity.

Owing to the fact that the ideal agent's behavioral policy is not affected by the variability of such experimental variables, it serves as a good proxy for assessing the degree of participants' engagement in model-based control. In principle, provided that the model-based agent has complete knowledge of the state-space and no cognitive constraints, it will always choose more optimally than a model-free agent.

To establish a clear link between behavior, experimental variables, and a computational model, we have added a new figure (Figure 2; see below). The figure shows that

- the choice optimality is clear behavioural readouts of each controller and is influenced by uncertainty and complexity (see our **reply 2-(3)** for more details), and
- the effect of those variables on choice behavior is well explained by key parameters of our model (see our **reply 2-(4)** for more details).

Behavioral readout of model-based control (computer simulation)

Behavioral readout of model-based control (subjects' data)

Figure 2. Behavioral results. (A) Choice optimality (a proxy for assessing the degree of agents' engagement in model-based control), of a model-based and model-free reinforcement learning agent. Choice optimality depicts the degree of match between agents' actual choices and an ideal agent's choice corrected for the number of available options. For full details of this measure, refer to Experimental Procedures. (B) Difference in choice optimality between a model-based and model-free reinforcement learning agent for the four experimental conditions (low/high state-transition uncertainty x low/high task complexity). Shown in red boxes are the effect of the two experimental variables on each measure (2-way repeated measures ANOVA). (C) Participants' choice optimality for the four experimental conditions. Shown in red boxes are the effect of the two experimental variables on each measure (2-way repeated measures ANOVA; also see Table S3 for full details). (D) Results of a general linear model analysis (dependent variable: choice optimality, independent variables: uncertainty, complexity, reward values, choices in the previous trial, and goal values). Uncertainty and complexity, the two key experimental variables in our task, significantly influence choice optimality (t-test; $p < 0.001$). Error bars are SEM across subjects.

- Second, depending on having only some outcomes available at a second state and still their value being instructed at the beginning of each trial, a model-based controller could make a specific choice for the first action by planning ahead to reach a certain outcome, thus, resulting in a specific measure of model-based control (I am not sure whether this is possible at all based on their design: so far, I believe not)

>> Reply 2-(2).

It is possible. Although we haven't made clear in our original manuscript (we apologize for this!), our task design does incorporate a specific behavioral marker, a choice bias, which could indicate goal-specific planning of the model-based controller. In the following sections, we show (i) how this behavioral measure is incorporated into the task design, (ii) behavioral results, and (iii) that this is reflected in valuation of the model.

(i) Left/right choice bias : a behavioral marker indicating reward-based learning and model-based control

Our task design involves an explicit manipulation of the goal. Note that the association between goal types and coin colors were randomized for each subject, and here we show one particular example (see the above figure). Let's define the L branch as the bottom left routes (accessible by making the L choice in the first stage) and the R branch as the top right routes (accessible by making the R choice in the first stage), respectively. The agent is not informed about task complexity until the second stage, so in the first stage a rational agent would make the following assumptions: in the first stage, outcome states associated with a silver coin are accessible by making a primary choice in both branches. An outcome state associated with a red coin are accessible by making a primary and a secondary choice in the L and R branch, respectively. An outcome state associated with a blue coin are accessible by making a secondary choice in both branches.

Accommodating this situation, we can roughly calculate the expected value of the L/R choice of the optimal agent (the same agent used to compute choice optimality) for the first stage. The probability of transitioning to a desired outcome state by making a primary and a secondary choice is given by (0.7, 0.3), which is computed by taking average of the two state-transition probability values: (0.9, 0.1) and (0.5, 0.5). Note that this setting is used to simulate average

experimental conditions. For the sake of simplicity, reward values were normalized to 1 (for the goal coin), 0.5 (for the other coins), and 0 (unrewarded).

If we assume that an agent relies on *model-free* control and that the agent makes a greedy choice, we can compute the expected values and the corresponding choice biases by using the uniform state-transition probability distribution (0.5,0.5) (meaning that the agent is agnostic about state-transition uncertainty and thus cannot afford to accommodate state-transition probability value changes) as follows.

- Silver coin : (the expected value of L branch) $1 \times 0.5 + 0 \times 0.5 = 0.5$. (the expected value of R branch) $1 \times 0.5 + 0.5 \times 0.5 = 0.75$. The expected value difference (L-R) = -0.25. We expect a R choice bias.
- Red coin : (the expected value of L branch) $1 \times 0.5 + 0.5 \times 0.5 = 0.75$. (the expected value of R branch) $0.5 \times 0.5 + 1 \times 0.5 = 0.75$. Therefore expected value difference (L-R) = 0. We expect no L choice bias.
- Blue coin : (the expected value of L branch) $0.5 \times 0.5 + 1 \times 0.5 = 0.75$. (the expected value of R branch) $0 \times 0.5 + 1 \times 0.5 = 0.5$. Therefore expected value difference (L-R) = +0.25. We expect L choice bias.

Therefore, if subjects performed the task using pure model-free control, they would show a well-balanced choice bias pattern: R bias, zero bias, and L bias for each goal, respectively.

On the other hand, if we assume that an agent relies on *model-based* control, we can compute the expected values and the corresponding choice biases, this time by using the average state-transition probability set (0.7,0.3) (meaning that the agent actively accommodates state-transition probability value changes between (0,9,0.1) and (0.5,0.5)) as follows.

- Silver coin : (the expected value of L branch) $1 \times \underline{0.7} + 0 \times \underline{0.3} = 0.7$. (the expected value of R branch) $1 \times \underline{0.7} + 0.5 \times \underline{0.3} = 0.85$. The expected value difference (L-R) = -0.15. We expect a weak R choice bias.
- Red coin : (the expected value of L branch) $1 \times \underline{0.7} + 0.5 \times \underline{0.3} = 0.85$. (the expected value of R branch) $0.5 \times \underline{0.7} + 1 \times \underline{0.3} = 0.65$. Therefore expected value difference (L-R) = +0.2. We expect a weak L choice bias.
- Blue coin : (the expected value of L branch) $0.5 \times \underline{0.7} + 1 \times \underline{0.3} = 0.65$. (the expected value of R branch) $0 \times \underline{0.7} + 1 \times \underline{0.3} = 0.3$. Therefore expected value difference (L-R) = +0.35. We expect L choice bias.

Therefore, if subjects performed the task using model-based control, they would exhibit a slight left bias pattern: weak R bias, weak L bias, and L bias for each goal, respectively.

The below figure illustrates these two cases.

(ii) Behavioral results

We found that the above-mentioned prediction about the choice bias matches subjects' actual choice bias (the left of the below figure). In particular, the data shows a clear left bias pattern, rejecting the null hypothesis that subjects used a pure model-free control strategy. This bias also affects choice consistency (the right of the below figure).

(iii) Model prediction

Furthermore, this choice bias and the corresponding choice consistency patterns are well predicted by the model (see below).

The following value difference pattern of the model further supports this finding, suggesting that these behavioral patterns are originated from the value learning.

Model's value difference (L-R)

Taken together, these results clearly indicate that subjects' choices are guided by reward-based learning, and further that their learning processes are guided by model-based control. We have added to the supplement, new figures that present these results (see Supplementary Figure S2 and Supplementary Methods).

A Behavioral readout of model-based control (ideal agent)

B Behavioral readout of value-based decision making (subjects' data)

C Behavioral readout of value-based decision making (computational model)

Figure S2. (A) Predicted choice bias patterns of model-free (MF) and model-based (MB) control, calculated for the three goal conditions defined as the trials according to which coin has the maximum monetary outcome value (low, medium, and high token value for the L choice). Owing to the asymmetric association between outcome states and coin types (For full details, see Supplementary Methods – Choice bias), participants would exhibit distinct choice bias patterns for each goal condition that distinguishes model-based from model-free control; the MF control agent would exhibit a balanced choice bias pattern, whereas the MB control agent would show a slight left bias pattern. For full details of this measure, refer to Supplementary Methods - Behavioral measure. **(B)** Participants' choice bias and choice consistency, the conventional behavioral markers indicating reward-based learning. Error bars are SEM across subjects. The prediction about the choice bias matches subjects' actual choice bias (the left of the below figure). In particular, the data shows a clear left bias pattern, rejecting the null hypothesis that subjects used a pure model-free control strategy. This bias is also reflected in choice consistency (the right plot). These results also indicate that participants' choice behavior is guided by reward-based learning more generally. **(C)** Choice bias (left), choice consistency (middle), and the average value difference (right) of our computational model of arbitration control (Figure 4B). For this, we ran a deterministic simulation in which our computational model experiences exactly the same episode of events as each individual subject, and we generated the trial-by-trial outputs. The max goal conditions are defined in the same way as in the Figure 2A. Error bars are SEM across subjects. Note that both the choice bias and choice consistency patterns of the model (the left and the middle plot) are fully consistent with the behavioral results (B). Second, the values difference (right – left choice) of the model is also consistent with this finding (the right plot), suggesting that these behavioral patterns are originated from value learning. In summary, our computational model encapsulates the essence of subjects' choice behavior as guided by reward-based learning.

“Behavioral measure (choice bias). In our task design, a left/right choice bias in the first stage can be interpreted as a behavioral marker indicating reward-based learning.

Our task design involves delicate manipulation of the goal. Note that the association between goal types and coin colors were randomized for each subject, and here we show one particular example (see the above figure). Let's define the R branch as the bottom left routes (accessible by making the R choice in the first stage) and the L branch as the top right routes (accessible by making the L choice in the first stage), respectively. The agent is not informed about task complexity until the second stage, so in the first stage a rational agent would make the following assumptions: in the first stage, outcome states associated with a silver coin are accessible by making a primary choice in both branches. An outcome state associated with a red coin are accessible by making a primary and a secondary choice in the R and L branch, respectively. An outcome state associated with a blue coin are accessible by making a secondary choice in both the R and the L branch.

Accommodating this situation, we can roughly calculate the expected value of the L/R choice of the optimal agent (the same agent used to compute choice optimality) for the first stage. The probability of transitioning to a desired outcome state by making a primary and a secondary choice is given by (0.7, 0.3), which is computed by taking average of the two state-transition probability values: (0.9, 0.1) and (0.5, 0.5). Note that this setting is used to simulate average experimental conditions. For the sake of simplicity, reward values were normalized to 1 (for the goal coin), 0.5 (for the other coins), and 0 (unrewarded).

If we assume that an agent relies on model-free control and that the agent makes a greedy choice, we can compute the expected values and the corresponding choice biases by using the uniform state-transition probability distribution (0.5,0.5) (meaning that the agent is agnostic about state-transition uncertainty and thus cannot afford to accommodate state-transition probability value changes) as follows. Note that the low, medium, and high value coin corresponds to silver, red, and blue coins in Figure 1 and Figure S1; again, the coin colors are randomized for each subject.

- Silver coin : (the expected value of L branch) $1 \times 0.5 + 0 \times 0.5 = 0.5$. (the expected value of R branch) $1 \times 0.5 + 0.5 \times 0.5 = 0.75$. The expected value difference (L-R) = -0.25. We expect a R choice bias.

- Red coin : (the expected value of L branch) $1 \times 0.5 + 0.5 \times 0.5 = 0.75$. (the expected value of R branch) $0.5 \times 0.5 + 1 \times 0.5 = 0.75$. Therefore expected value difference (L-R) = 0. We expect no L choice bias.

- Blue coin : (the expected value of L branch) $0.5 \times 0.5 + 1 \times 0.5 = 0.75$. (the expected value of R branch) $0 \times 0.5 + 1 \times 0.5 = 0.5$. Therefore expected value difference (L-R) = +0.25. We expect L choice bias.

Therefore, if subjects performed the task using pure model-free control, they would show a well-balanced choice bias pattern: R bias, zero bias, and L bias for each goal, respectively.

On the other hand, if we assume that an agent relies on model-based control, we can compute the expected values and the corresponding choice biases, this time by using the average state-transition probability set (0.7,0.3) (meaning that the agent actively accommodates state-transition probability value changes between (0.9,0.1) and (0.5,0.5)) as follows.

- Silver coin : (the expected value of L branch) $1 \times 0.7 + 0 \times 0.3 = 0.7$. (the expected value of R branch) $1 \times 0.7 + 0.5 \times 0.3 = 0.85$. The expected value difference (L-R) = -0.15. We expect a weak R choice bias.

- Red coin : (the expected value of L branch) $1 \times 0.7 + 0.5 \times 0.3 = 0.85$. (the expected value of R branch) $0.5 \times 0.7 + 1 \times 0.3 = 0.65$. Therefore expected value difference (L-R) = +0.2. We expect a weak L choice bias.

- Blue coin : (the expected value of L branch) $0.5 \times 0.7 + 1 \times 0.3 = 0.65$. (the expected value of R branch) $0 \times 0.7 + 1 \times 0.3 = 0.3$. Therefore expected value difference (L-R) = +0.35. We expect L choice bias.

Therefore, if subjects performed the task using model-based control, they would exhibit a slight left bias pattern: weak R bias, weak L bias, and L bias for each goal, respectively.”

- Having established clear behavioural readouts of each controller (e.g. in logistic regression model on choice repetition), one can test straightforward the influence of uncertainty and complexity on these.

>> Reply 2-(3).

Since we have established a clear behavioral measure to read out model-based/model-free control (choice optimality; reply 2-(1)), we have tested the influence of the two experimental variables (uncertainty and complexity) on the choice optimality.

(i) Influence of uncertainty and complexity on choice optimality

For this, we ran a GLM analysis with the choice optimality being included as a dependent variable, and uncertainty, complexity, reward values, choices in the previous trial, and goal values as independent variables. The figure shown below is the beta estimates of the GLM analysis. We found that both uncertainty and complexity significantly affected to choice optimality (t-test $p < 0.001$); influence of the other three variables on choice optimality were not significant.

Note that this is fully consistent with our behavioral results (below).

(ii) Uncertainty and complexity effect on choice optimality are explained by the model

Notably, when having the best fitting model perform the task, we found that the model exhibits choice optimality patterns similar to subjects.

To further compare the degree of influence of uncertainty and complexity on choice optimality in subjects' behavior with model's prediction, we ran the same type of the GLM analysis as in (i) on model's behavioral data, and compared this effect with the effect on actual subjects' behavioral data. The model's behavioral data was generated by having our model perform the task, the same task previously used to collect each individual subject's behavioral data.

We found a significant correlation between the effect sizes of these two cases (see the figure below; it is included as Figure 5C and 5D in the revised version of our manuscript), suggesting that our model encapsulates the essence of behavior guided by model-based control.

Effect of uncertainty and complexity on model's choice optimality (Behavior recovery analysis)

Note that our model, which incorporates both uncertainty and complexity, accounts for this effect significantly better than alternative models including a pure model-free version, a pure model-based version, and the Lee2014 model that incorporates only the effects of uncertainty on arbitration (see below; Supplementary Figure S6B and S6C in the revised version of our manuscript).

Uncertainty effect (comparison between behavioral and simulation data)

Complexity effect (comparison between behavioral and simulation data)

These results clearly suggest that our model encapsulates the essence of subjects' choice behavior associated with uncertainty and complexity significantly better than the alternative model that does not accommodate complexity.

In summary, we have shown in Section (i) that manipulation of the two task variables successfully influences the dimension of choice behavior that can read out model-based/model-free control, and in Section (ii) that the behavioral patterns associated with this effect is well encapsulated by the model.

As a result, the figures produced in our reply 2-(2) and 2-(3) are included in Figure 2 and Figure 5 of the revised version of the manuscript respectively (reproduced below).

Behavioral readout of model-based control (computer simulation)

Behavioral readout of model-based control (subjects' data)

Figure 2. Behavioral results. (A) Choice optimality (a proxy for assessing the degree of agents' engagement in model-based control), of a model-based and model-free reinforcement learning agent. Choice optimality depicts the degree of match between agents' actual choices and an ideal agent's choice corrected for the number of available options. For full details of this measure, refer to Experimental Procedures. **(B)** Difference in choice optimality between a model-based and model-free reinforcement learning agent for the four experimental conditions (low/high state-transition uncertainty x low/high task complexity). Shown in red boxes are the effect of the two experimental variables on each measure (2-way repeated measures ANOVA). **(C)** Participants' choice optimality for the four experimental conditions. Shown in red boxes are the effect of the two experimental variables on each measure (2-way repeated measures ANOVA; also see Table S3 for full details). **(D)** Results of a general linear model analysis (dependent variable: choice optimality, independent variables: uncertainty, complexity, reward values, choices in the previous trial, and goal values). Uncertainty and complexity, the two key experimental variables in our task, significantly influence choice optimality (t-test; $p < 0.001$). Error bars are SEM across subjects.

Behavioral readout of model-based control (computational model)

Effect of uncertainty and complexity on model's choice optimality (Behavior recovery analysis)

Figure 5. Computational model fitting results. (A) Patterns of choice optimality generated by the best fitting version of the arbitration model, using parameters obtained from fitting to participants behavior. For this, the model was run on the task (1000 times), and we computed choice optimality measures in the same way as in Fig. 2B and 2C. (B) Degree of engagement of model-based control predicted by the computational model, based on the model fits to individual participants. P_{MB} corresponds to the weights allocated to the MB strategy. Shown in the red box are the effect of the two experimental variables on each measure (2-way repeated measures ANOVA; also see Table S4 for full details). Error bars are SEM across subjects. (C, D) Behavioral effect recovery analysis. The individual effect sizes of uncertainty (C) and complexity (D) on choice optimality of subjects (true data) were compared with those of our computational model (simulated data).

- Subsequently, making very clear, which parameter(s) in their model(s) influence(s) what kind of behaviour (and how) seems necessary and it should be straightforward to be demonstrated in simulations.

>> Reply 2-(4).

To establish a link between model parameters and behavior patterns, we ran a parameter recovery analysis.

The parameter recovery analysis evaluates the degree of consistency between data-to-model parameter and model parameter-to-data conversion. It consists of the following sequence of processes: Subjects' data → model fitting 1 (original parameters) → simulated data → model fitting 2 (recovered parameters). The simulated data were generated by running simulations with the best fitting model on the original task.

If the model parameter(s) successfully encoded the key features of behavior, then the corresponding parameters should be well recovered. In other words, the original and the recovered parameter values are significantly correlated. If the model's predictions are based on less important dimensions of behavior (e.g., overfitting), then the principal dimensions of the simulated data would likely diverge from those of the original subjects' data, leading to unsuccessful recovery of parameter values.

Here we show that the four key parameters of the model (the sensitivity of reward and state prediction error, learning rates for model-based and model-free control, exploration sensitivity) are necessary for accounting for subjects' behavior patterns. We found significant correlations between the original and the recovered parameter values (see the figure below), highlighting the necessity of the key parameters of the model, as well as helping to rule out a substantive effect of overfitting.

Recoverability of the computational model's key parameters

This result has been included in Supplementary Information (Figure S5), and the discussion has been added to the main text (see below).

“The best fitting model we identified in the previous section encapsulates the extent to which participants' choice behavior is guided by reward-based learning more generally (Figure S2). Note that this measure also allows us to distinguish model-based control from model-free control (Figure S2A), and we found evidence in subjects' data to fully dismiss the possibility that subjects use a pure model-free control strategy (Figure S2B). In addition, we ran a parameter recovery analysis to further establish a link between choice behavior and the computations underlying arbitration control, and found that the model's key parameters were successfully recovered from the behavior of the best fitting model (Parameter recovery analysis; Figure S5).”

- The “high uncertainty condition” with 50/50 transitions remains conceptually unclear to me. While it is obvious that this induces high levels of uncertainty and reduces model-based control,

the authors mention themselves that this results in a random transition, thus, rendering any kind of model-based control meaningless because there is essentially no structure to be detected. I can see why this is under certain circumstances an interesting condition to be included but think that for demonstrating their argument using an additional shift from 90/10 to 70/30 or 60/40 would have been more informative because model-based behaviour (if distinctly detectable with this task at all) would still have been possible

>> Reply 2-(5).

Thanks for sharing valuable insights with us. The reviewer is correct in that in a 50/50 transition condition model-based control is nullified in the long run, provided that the model-based learner fully learned this state-transition probability values. With an ideal learner capable of precisely and quickly tracking state-transition probability changes, it would be good to switch between 90/10 (low uncertainty) and 70/30 (medium uncertainty). However, from our previous study (Lee et al., 2014) and following simulation analyses, we learned that switching between 90/10 (low uncertainty) and 50/50 (high uncertainty) is a better way to maximize variability in the amount of SPE. This condition is thus intended to elicit a large amount of SPE.

We agree that this point should be made clear to preclude any misunderstanding, so we have elaborated this point in both the main text and the Experimental Procedures section.

“Note that our high uncertainty condition is intended to maximize variability in the amount of SPE, as opposed to making participants perfectly learn the state-transition probabilities (0.5, 0.5). In fact, it would be more challenging to test for effects on behavior of other more moderate uncertainty conditions, such as (0.7, 0.3) or (0.6, 0.4), within relatively short blocks of trials.”

“... it was previously shown that with (0.9,0.1) participants feel that the state transition is congruent with the choice, whereas with (0.5,0.5) the state transition is random (Lee et al., 2014). Furthermore, the changes at these rates ensures that tonically varying changes in model-based vs model-free control can be detected at experimental frequencies appropriate for fMRI data. ”

- The way how the exact number (or range) of trials per block and per condition was determined cannot be clearly followed from the manuscript. Please specify in a way that other researcher could reproduce the task based on the manuscript

>> Reply 2-(6). Thank you for the suggestion. We have supplemented the Experimental Procedures section, which provides full details for task implementation including trial ranges per blocks/conditions. In addition, we have make the stimulus program (based on a MATLAB psychtoolbox) available to download through our GitHub link (refer to Data and Software Availability section), so that other researcher can easily reproduce and revise our task.

If the authors can show clear distinct behavioural readouts for model-free and model-based control in this task (or could present a stringent and thus convincing argument why they feel this

is not necessary), they still need to demonstrate whether their inferred parameters can actually recover the key behavioural features of their task. This is essentially necessary.

>> **Reply 2-(7)**. This has been fully resolved in addressing the reviewer's previous comment. Please refer to the **reply 2-(4)**.

Please also include the per-subject negative log-likelihood to the supplemental table showing model parameters and please use histograms or a table with percentiles to show distribution of parameters and the likelihood (and there measure of model evidence). Although it is very much appreciated that the authors share behavioural data and code, there is no word on how they infer parameters from the behavioural data and how they approximate or estimate the log model evidence, which they have to enter in the RFX BMS.

>> **Reply 2-(8)**. We have updated Supplementary Table S1 showing all the parameter values and likelihood values for each subject, and added a histogram plot (Figure S3) to show the parameter distribution. We also have updated the Experimental procedures section as requested by the reviewer.

Likelihood value was calculated by using the model's softmax function of the action taken by each subject. Free parameters were optimized to minimize the sum of negative log likelihood using a Nelder-Mead simplex algorithm. For more details, please refer to the supplementary document of Lee et al. (Neuron, 2014). The section "Parameter Estimation" in supplementary methods provides all the details for inferring parameters.

As mentioned before, the fMRI results could be very interesting but I refrain to comment on them because all fMRI analyses crucially rest upon regressors extracted from the model.

I hope the authors can find these comments helpful as they are not meant to devalue their work. However, I feel it is necessary to present clear behavioural readouts when talking about model-free and model-based control and their arbitration. I believe the authors might agree on this perspective. Distinct behavioural readouts have already been challenging to understand in their previous paper from 2014.

>> **Reply 2-(9)**. Again, we greatly appreciate the reviewer's insightful comments. As shown in our replies above, we have fully addressed all the issues, including behavioral readouts, its relationships to model-based and model-free control, and additional validations using the parameter recovery analysis. We thus believe that they truly justify our neural analysis.

Reviewer #3 (Remarks to the Author):

Kim et al. present a study that builds on previous work from Lee and O'Doherty investigating the arbitration between putative model-free and model-based controllers. In this study they additionally modulate the "complexity" of the task, by manipulating the number of choice options (low: 2 vs high: 4) at the second choice stage of the task. This leads to a 2x2 factorial design in which state uncertainty (low: 0.5/0.5 vs high: 0.9/0.1) is crossed with complexity. Behaviorally they show that somewhat counterintuitively, increased complexity (defined in this way) leads to more model-based control, whereas increased uncertainty leads to more model-free control, as expected. Furthermore, they show that there is a logical interaction which shows that with greater uncertainty the effect of complexity is attenuated, unsurprisingly. This is an interesting pattern of behavioral patterns that adds to the field's understanding of these putative control systems. In their fMRI results, they first nicely replicate their previous work showing reliability effects in bilateral IIPFC and integrated value signals (which in effect reflect a comparison between model-based and model-free controllers) in vmPFC. They then show that a) their new model that incorporates task complexity better accounts for the BOLD signals in these areas; and b) there is an interaction with task complexity, although I do have a query about its interpretation. Overall, this is an impressive, rigorous study that advances our understanding of the conditions under which model-based versus model-free systems win out to control behavior, a topic likely to be of interest to a broad audience.

More details of the task could be provided and I have a few questions about the task/model:

How often does the task transition between high and low states of uncertainty? Do the interaction effects depend on whether this has just happened or many trials earlier (in particular for the transition from high to low uncertainty)?

>> Reply 3-(1).

The transitions between the low and the high uncertainty block takes place about every 15 trials (13-17 trials). Note that the length of trials is determined to ensure that the estimation process of the state-transition probability of the model-based learner does not break down.

To answer the reviewer's question, we examined the interaction effects on choice behavior (choice optimality) and model preference (probability of choosing model-based control) in the first and the last half of each block, respectively. Despite insufficient sample sizes leading to a low statistical power, the interaction effects appear to exist in both cases (see below results). The effect seems to be a bit stronger for the late half of the blocks (F score = 6.64 (early) and 19.73 (late) for choice optimality; F score = 3.08 (early) and 3.98 (late) for model preference), suggesting a possibility that the interaction effects arises some time after the condition changes. This makes perfect sense because the uncertainty change is recognizable only through a change in the average amount of state prediction error.

Tests of Within-Subjects Contrasts

Measure: MEASURE_1

Source	Uncertainty	Complexity	Type III Sum of Squares	df	Mean Square	F	Sig.
Uncertainty	Linear		.186	1	.186	29.247	.000
Error(Uncertainty)	Linear		.146	23	.006		
Complexity		Linear	.080	1	.080	17.364	.000
Error(Complexity)		Linear	.106	23	.005		
Uncertainty * Complexity	Linear	Linear	.021	1	.021	6.638	.017
Error (Uncertainty*Complexity)	Linear	Linear	.071	23	.003		

Figure. Effect of the uncertainty and complexity (experimental conditions) on choice optimality, computed by using the 1st-50th percentile of trials in each block

Tests of Within-Subjects Contrasts

Measure: MEASURE_1

Source	Uncertainty	Complexity	Type III Sum of Squares	df	Mean Square	F	Sig.
Uncertainty	Linear		.209	1	.209	34.465	.000
Error(Uncertainty)	Linear		.139	23	.006		
Complexity		Linear	.113	1	.113	29.175	.000
Error(Complexity)		Linear	.089	23	.004		
Uncertainty * Complexity	Linear	Linear	.073	1	.073	19.733	.000
Error (Uncertainty*Complexity)	Linear	Linear	.085	23	.004		

Figure. Effect of the uncertainty and complexity (experimental conditions) on choice optimality, computed by using the 51st-100th percentile of trials in each block

Tests of Within-Subjects Contrasts

Measure: MEASURE_1

Source	Uncertainty	Complexity	Type III Sum of Squares	df	Mean Square	F	Sig.
Uncertainty	Linear		.000	1	.000	.342	.565
Error(Uncertainty)	Linear		.015	23	.001		
Complexity		Linear	.110	1	.110	87.862	.000
Error(Complexity)		Linear	.029	23	.001		
Uncertainty * Complexity	Linear	Linear	.003	1	.003	3.084	.092
Error (Uncertainty*Complexity)	Linear	Linear	.025	23	.001		

Figure. Effect of the uncertainty and complexity (experimental conditions) on model preference, computed by using the 1st-50th percentile of trials in each block

Measure: MEASURE_1

Tests of Within-Subjects Contrasts

Source	Uncertainty	Complexity	Type III Sum of Squares	df	Mean Square	F	Sig.
Uncertainty	Linear		.024	1	.024	46.966	.000
Error(Uncertainty)	Linear		.012	23	.001		
Complexity		Linear	.138	1	.138	138.123	.000
Error(Complexity)		Linear	.023	23	.001		
Uncertainty * Complexity	Linear	Linear	.003	1	.003	3.983	.058
Error (Uncertainty*Complexity)	Linear	Linear	.016	23	.001		

Figure. Effect of the uncertainty and complexity (experimental conditions) on model preference, computed by using the 51st-100th percentile of trials in each block

We were not able to specifically assess the effect for the transition from high to low uncertainty due to the lack of a sufficient number of trials for this particular comparison. Testing for this would require a modification of our task to provide a clear contrast between low-to-high vs high-to-low uncertainty transitions, which is an experimental manipulation that we must leave for future studies.

How is the goal change regressor defined?

>> **Reply 3-(2).** We use a max rule to define a goal, that is, on each trial the goal is set to the token associated with the highest value. The goal change regressor has the value 1 if the goal of the current trial is different from that of the previous trial. For example,

- Token values (silver, red, blue) : (7,3,5), (2,9,5), and (3,7,3) in trial 1,2,3, respectively.
- Goal : silver, red, and red in trial 1,2,3, respectively.
- Goal change regressor : 0,1,0 in trial 1,2,3, respectively.

Clarification is provided in Supplementary Methods - GLM design.

“GLM design. A general linear model (GLM) was used to generate voxelwise statistical parametric maps (SPMs) from the fMRI data. We created subject-specific design matrices containing the following regressors:

(R1) regressors encoding the average BOLD response at two choice states and one outcome states, (R2,R3) two parametric regressors encoding the model-derived prediction error signals – state prediction error (SPE) of MB and reward prediction error (RPE) of MF, (R4) a regressor encoding the average BOLD response at the start of each choice state (the time of presentation of the values of each token in the first stage and the time of the state presentation in the second stage), (R5) a parametric regressor encoding the goal change; it is a binary variable indicating whether the type of a coin associated with the largest value is different from the one in the previous trial.... (the rest is omitted)”

I could not work it out but does not task complexity influence the computation of reliability in the model, since higher number of choice options should increase the entropy of the choice? Is this relation already taken into consideration in the computation of reliability in the model?

>> **Reply 3-(3).** The reliability computation itself does not take into account choice entropy in our computational model. Our model is designed in a way that separately computes reliability based on prediction error and task complexity based on the number of choices (choice entropy), and then these two signals are mixed to arbitrate between model-based and model-free control (please refer to Figure 3 for the model structure and Figure 8 for the neural effect). That being said, we cannot rule out the possibility that choice entropy influences the sensitivity prediction error, which should be left for further study.

Figure 7 suggests that the iIPFC effect goes in the opposite direction of the behavioral results. When complexity is higher, the MB reliability signal is lower, and vice versa? How should we conceptualize this interaction then?

>> **Reply 3-(4).**

Figure 7 (that became Figure 8 in the revised version), shows that changes in the correlation between activity in vIPFC and reliability occur as a function of changes in complexity for both MB and MF reliability signals. According to the arbitration theory, choice behavior is determined through an interaction between MB and MF reliability and complexity. Given the complex nature of this three-way interaction, it is hard to conceptualize the interaction as it is plotted. It is also important to take into account, that an additional factor that we think would be important for understanding the pattern of vIPFC activity, is that according to our hypothesis about vIPFC's role in arbitration, we suspect that this region is acting to inhibit MF signals elsewhere, when MB control is required. Thus, on top of the reliability and complexity signals, we may expect to see changes in vIPFC activity that reflects the act of exerting inhibitory control on the MF system when MB control is required. Thus, the expected behavior of the signals in this region is very complicated indeed and may not lend itself to a straightforward interpretation. We think that what is really needed is a more mechanistic or implementational level model of vIPFC function in order to make clearer theoretical predictions of what this region is doing in arbitration. The model we have now is essentially an algorithmic or process level model -- it doesn't say much about how these processes are actually likely to be implemented in the brain during the arbitration process. Developing such an implementational level model is beyond the scope of the current manuscript, and honestly, may be beyond our reach for some time until we have much better characterized and measured the neuronal dynamics in vIPFC, likely with more refined methods than fMRI (such as single-unit electrophysiology). For now, we think that the best we can say is that there is an interaction between these variables in this region which is consistent with each of these variables being utilized in the arbitration process. Given these issues, we prefer to leave this as a limitation of this study, specifically by pointing out that a further study needs to use techniques with a better resolution to understand the nature of this computation.

Behaviorally the results show that somewhat counterintuitively, increased complexity leads to more model-based control. This suggests that having to plan more carefully, or to consider more options, actually makes people more model-based. A suggestion I hope will be helpful: it would be interesting to explore at what point this breaks down due to over-taxed cognitive demands.

>> Reply 3-(5).

Indeed, this is an important question. As we have shown in the manuscript, our results do at least to some degree shed light on this question, because we have shown that when state-space uncertainty is high AND complexity is high, indeed participants become LESS model-based and more model-free, suggesting that increased recruitment of model-based based on increased task demands does eventually break down (as was our original hypothesis). We agree that it will be important to further investigate this phenomenon in future studies to investigate more comprehensively the conditions under which increased MB control breaks down (such that participants essentially give up). We also suspect that individual differences such as IQ and WM capacity are going to play a key moderating role in this process.

We have added the following discussion point on this topic to the paper:

“Another way of interpreting these findings is that when task demands increase but yet the MB system is capable of meeting those challenges, then MB control can and does step up to meet the challenge, but if task demands get too difficult, potentially beyond the capacity of the MB system, then MF control takes over by default. It is likely that individual differences in executive function such as working memory capacity will moderate this effect across participants, as has been shown in the case of other challenges to MB control such as stress induction (Otto et al., 2013; Quaedflieg et al 2019).”

We greatly appreciate the reviewer’s very insightful and constructive comments, which have made it possible for us to significantly improve our paper.

Reviewers' comments:

Reviewer #1 (Remarks to the Author):

The authors have adequately addressed my concerns about computational cost, clarifying new vs. replicated findings and Bayesian model selection.

As it stands, I still have major concerns about the conclusions that can be drawn from the findings. The entire manuscript hinges on choice optimality, which is an independent behavioral measure quantifying how well the subject's matches that of an ideal MB agent. Indeed, all neural signatures of interest to the authors relate to choice optimality, not choice behavior. Yet, at no point are we given a complete description of choice behavior used to generate choice optimality measure.

The authors now include some basic, descriptive data about the subject's performance in the task. Figure S2B shows data for L/R choice in stage 1 as well as a measure of choice consistency in stage 2. However, the behavioral data shown in Figure S2B are collapsed across all conditions of complexity and uncertainty. The authors argue this is valid for the stage 1 decision because complexity is not introduced until stage 2. But rather than speculate or make assumptions, behavior could be quantified. This begins to give us an idea of the pattern of choice indicative of MB control, and therefore choice optimality, but this is only a start.

A full analysis must include descriptions of subject choice (number, percentage, and latency) at each stage for each of the four conditions. For example, when choice optimality is high vs low, what does choice behavior look like in the four separate conditions? Is it identical? Independent of choice optimality, how do uncertainty and complexity affect reaction time and choice behavior in each of the four conditions?

Further, what is the relationship between the observed neural signatures of arbitration and the subject's actual choice behavior? This is not trivial. Throughout the discussion, the authors link neural processes of arbitration to behavior.

These are just excerpts from first paragraph:

"These behavioral findings were supported by evidence that a region of the brain previously implicated in the arbitration process..."

"Taken together, these findings help to advance our understanding of the contribution of two key variables to the arbitration process at behavioral and neural levels."

Yet, the authors have only actually linked neural signatures to choice optimality, which asks if the choice behavior pattern matches that of the ideal MB agent. MB and MF strategies are ultimately just constructs that allow us to explain patterns of choice behavior. Linking neural signatures of arbitration to actual choice behavior is key to the goals of this study. As it stands, this link has not conclusively been made.

Reviewer #2 (Remarks to the Author):

Overall, the authors performed a thorough revision and were responsive to the critique raised. I do have a few rather minor comments.

Re Reply 2-(1), (i) Remarks on choice repetition: At the end of their answer the authors conclude 'Thus in our study, the application of the choice consistency measure is confined to a simple readout of reward-based value learning guided by model-free control'. This is actually what I asked for ('which could constitute a measure of model-free control'). I do see why measures of choice repetition / consistency are not well applicable for model-based control and also do not to indicate the arbitration between MF and MB. Thus, I do appreciate the authors' presentation of choice

optimality. The reports of choice optimality are overall convincing for indicating the superiority of model-based control in this particular environment and the effect of the manipulations that do affect on arbitration. However, it does not provide a distinct behavioural readout because it assumes that one strategy is simply more optimal than the other which substantially relies upon how the environment is designed. You could also design a task the way that MF would be more optimal. However, for some measure of model-free control, I think it is possible to demonstrate it as outlined before (and the authors actually do that in the supplement). Also by using simulations of a model-free learner only, it should be possible to demonstrate this even clearer (once again this seems to be done in the supplement now). I think the supplemental analysis presented in S-Figure 2 is quite interesting and should be moved to the main manuscript (choice bias and choice consistency). I find it an important demonstration of the behaviour going on in this task. To be able to incorporate this, the choice optimality section could be shortened a bit or parts be moved to the supplement (it is a bit lengthy at some point).

Behavior and parameter recovery make an important contribution. I value them much more than the large scale model comparison.

For the fMRI analysis, I think some more specification of the GLM needs to go to the main manuscript.

“Finally, we implemented a second-level random effects analysis for each regressor of interest...” what does that mean exactly? Where they all included in one second-level random effects model or each in a separate? Please specify and explain the reasoning.

Reviewer #3 (Remarks to the Author):

The authors have comprehensively addressed my comments and I have no further concerns. I would like to commend them on a very nice paper.

Reviewers' comments:

We thank the reviewers for their comments. Below we detail our response to the reviewer comments.

Please also note one other change to the figures was made in the course of this revision. In the original Figure 2D and Figure S6A, the error bars had depicted the standard deviation instead of the standard error of the mean (SEM). We now changed these plots to show the SEM, so as to use a consistent measure across all figures in the manuscript.

Reviewer #1 (Remarks to the Author):

The authors have adequately addressed my concerns about computational cost, clarifying new vs. replicated findings and Bayesian model selection.

As it stands, I still have major concerns about the conclusions that can be drawn from the findings. The entire manuscript hinges on choice optimality, which is an independent behavioral measure quantifying how well the subject's matches that of an ideal MB agent. Indeed, all neural signatures of interest to the authors relate to choice optimality, not choice behavior. Yet, at no point are we given a complete description of choice behavior used to generate choice optimality measure.

>> [Reply 1-1]

We appreciate the reviewer's insightful comments about the relationship between choice behavior and choice optimality. We believe that choice optimality is a sensible measure that can provide a model independent behavioral profile of model-based control on this particular task. We disagree with the reviewer that "the entire manuscript hinges on choice optimality". Instead, choice optimality simply complements the model-driven analyses by providing a model independent measure of model-based behavior. That being said, we agree with the reviewer that a more detailed description of choice behavior potentially associated with the choice optimality would make our findings more convincing.

1. Choice switching is contingent on goal change.

To fully reflect the reviewer's concern, we examined choice behavior in a situation in which subjects need to set a new goal. A goal change necessitates a change in strategy, and the degree to which people switch their strategy would relate to the extent to which they are engaging model-based control. Thus, we argue that choice switching associated with goal change is a possible basis for choice optimality. Indeed, it is sensitive to goal change (see the figure below).

Figure. Subjects' ratio of choice switching in the second stage following goal change vs no goal change. (paired t-test; left $p = 1.2e-4$). Note that this measure is valid for only the second stage because in the first stage, each of the choices are optimal in half of the trials (in the high uncertainty condition). The goal-stay condition refers to the trials in which the *same* token values are being used on trial t and trial $t+1$ or at least

token values even if different that would promote the same optimal choice on the proceeding compared to the next trial. The goal-change condition refers to the situation where the change in token values from trial t to trial $t+1$ necessitates a *change* in choice behavior from trial t to $t+1$.

2. Choice switching is sensitive to experimental conditions.

To see if this choice behavior is affected by the experimental manipulation, we then examined the ratio of choice switching separately for each level of uncertainty and complexity. We found both a significant the main and interaction effect of uncertainty and complexity on choice switching (see the below figure and the result of two-way repeated measure ANOVA). The effect patterns are mostly consistent with the ones with choice optimality.

Tests of Within-Subjects Contrasts

Measure: MEASURE_1

Source	uncertainty	complexity	Type III Sum of Squares	df	Mean Square	F	Sig.
uncertainty	Linear		.420	1	.420	90.044	.000
Error(uncertainty)	Linear		.107	23	.005		
complexity		Linear	.420	1	.420	90.044	.000
Error(complexity)		Linear	.107	23	.005		
uncertainty * complexity	Linear	Linear	.227	1	.227	27.675	.000
Error (uncertainty*complexity)	Linear	Linear	.188	23	.008		

Note however that the pattern of choice switching shown above does not completely align with choice optimality. This can be seen for the case of high uncertainty where choice switching frequency increases relative to the low uncertainty conditions, whereas a decrease in choice optimality occurs in high uncertainty relative to low uncertainty conditions. This difference can be explained by the fact that switching choice can lead to subsequent choice of a good option or a poor option. To do the task effectively, one needs to switch to a good option (not merely increase switching rate per se). In the high uncertainty conditions we found that switching to the objectively better choice is indeed significantly reduced in this condition, compared to the low uncertainty conditions ($p < 0.01$; paired t-test). We also found that subjects' earning ratio (= actual reward / maximum possible reward in each trial) is significantly reduced in the high uncertainty conditions relative to the other conditions ($p < 0.001$; paired t-test). Taken together these findings provide an explanation for the behavioral underpinnings of the choice optimality measure reported in Figure 3C (in the revised version of our manuscript).

3. Link between choice switching and model-based control

We further investigated whether goal-change driven choice switching behavior can be diagnostic of model-based control. For this, we examined the ratio of choice switching is different as a function of the degree of model-based control (PMB : the probability of choosing model-based strategy). We found both a significant main and interaction effect of model-based control (PMB) and goal change on choice switching (see the below figure and the result of two-way repeated measure ANOVA).

Tests of Within-Subjects Contrasts

Measure: MEASURE_1

Source	maxgoal	pmb	Type III Sum of Squares	df	Mean Square	F	Sig.
maxgoal	Linear		.197	1	.197	18.885	.000
Error(maxgoal)	Linear		.239	23	.010		
pmb		Linear	.322	1	.322	44.500	.000
Error(pmb)		Linear	.167	23	.007		
maxgoal * pmb	Linear	Linear	.096	1	.096	13.169	.001
Error(maxgoal*pmb)	Linear	Linear	.168	23	.007		

4. Link between choice switching and choice optimality measure

Finally, in order to establish a direct link between choice switching and choice optimality, our behavioral measure indicating model-based control, we quantified the average correlation for each participant between choice optimality and the choice switching ratio. Specifically, we ran GLM analyses to compute an effect size for each individual subject, and found that all of the individual effect sizes are positive. (see below)

Effect of choice switch on choice optimality

Y: choice switch, X: choice optimality

Mean = 0.6007; $p=1.1102e-16$

Effect of choice optimality on choice switch

Y: choice switch, X: choice optimality

Mean = 0.43416; $p=3.3307e-16$

We obtained the same result with the logistic regression analysis:

$$\ln(\text{switch}/\text{no switch}) = \text{constant} + \text{beta1} * \text{choice optimality}$$

Mean(effect size) = 0.57772
 $P=1.656e-08$

In summary, we formally established a link between the experimental manipulations (goal changes,

uncertainty, and complexity), subjects' choice behavior (choice switching), subjects' choice optimality, and subjects' learning strategy (model-based control).

These results have been included in the supplementary information (Figure S2) and discussions have been added to the main text (see below).

Figure S2. Choice switching. (A) Choice switching is contingent on goal change. We examined choice behavior in a situation in which subjects need to set a new goal. A goal change necessitates a change in strategy, the degree to which people switch their strategy would relate to the extent to which they are engaging model-based control. Shown are subjects' ratio of choice switching in the second stage following goal change vs no goal change. (paired t -test; left $p = 1.2 \times 10^{-4}$). Note that this measure is valid for only the second stage because in the first stage, each of the choices are optimal in half of the trials (in the high uncertainty condition). The goal-stay condition refers to the trials in which the *same* token values are being used on trial t and trial $t+1$ or at least token values even if different that would promote the same optimal choice on the proceeding compared to the next trial. The goal-change condition refers to the situation where the change in token values from trial t to trial $t+1$ necessitates a *change* in choice behavior from trial t to $t+1$. **(B)** Choice switching is sensitive to experimental conditions. To see if this choice behavior is affected by the experimental manipulation, we then examined the ratio of choice switching separately for each level of uncertainty and complexity. We found both a significant main and interaction effect of uncertainty and complexity on choice switching (two-way repeated measure ANOVA; $p < 0.001$ for both main and interaction effects). The effect patterns are mostly consistent with that for choice optimality. Note however that the pattern of choice switching shown above does not completely align with choice optimality. This can be seen for the case of high uncertainty where choice switching frequency increases relative to the low uncertainty conditions, whereas a decrease in choice optimality occurs in high uncertainty relative to low uncertainty conditions. This difference can be explained by the fact that switching choice can lead to subsequent choice of a good option or a poor option. To do the task effectively, one needs to switch to a good option (not merely increase switching rate per se). In the high uncertainty conditions we found that switching to the objectively better choice is indeed significantly reduced in this condition, compared to the low uncertainty conditions ($p < 0.01$; paired t -test). We also found that subjects' earning ratio (= actual reward / maximum possible reward in each trial) is significantly reduced in the high uncertainty conditions relative to the other conditions ($p < 0.001$; paired t -test). Taken together these findings provide an explanation for the behavioral underpinnings of the choice optimality measure reported in Figure 3C. **(C)** Link between choice switching and model-based control. We further investigated whether choice switching behavior can be diagnostic of model-based control. For this, we examined the ratio of choice switching is different as a function of the degree of model-based control (PMB : the probability of choosing model-based strategy). We found both a significant main and interaction effect of model-based control (PMB) and goal change on choice switching (two-way repeated measure ANOVA; $p < 0.001$ for the main effects and $p = 0.001$ for the interaction effect). **(D)** Link between choice switching and choice optimality. Finally, in order to establish the relationship between choice switching and choice optimality, our behavioral measure indicating model-based control, we quantified the average correlation for each participant between choice optimality and the choice switching ratio. Specifically, we ran GLM analyses to compute an effect size for each individual subject, and found that all of the individual effect sizes are positive (mean effect size = 0.6; $p = 1.1 \times 10^{-16}$). We obtained the same result with the logistic regression analysis (mean effect size = 0.57; $p = 1.7 \times 10^{-8}$) Taken together, our findings help establish a link between the experimental

manipulations (goal changes, uncertainty, and complexity), and the participants' choice behavior (choice switching), choice optimality, and learning strategy (model-based control). All error bars are SEM across subjects.

Main text:

“Although choice optimality provides a model independent behavioral profile of model-based control on this particular task, an open question concerns what specific patterns of choice behavior produces high choice optimality. To shed light on this, we focused on choice behavior after a change in the token values has occurred that would necessitate a change in the goal compared to the previous trial. The degree to which people switch their strategy in response to the need to change goal should relate to the extent to which they are engaging model-based control. Consistent with this possibility, we found that choice switching associated with goal change (in combination with also choosing a better alternative on the next trial) provides a good account of choice optimality (Figure S2).

In summary, we formally established a link between the experimental manipulations (goal changes, uncertainty, and complexity), subjects' choice behavior (choice switching), subjects' choice optimality, and subjects' learning strategy (model-based control).”

5. Effects of task conditions on choice latency.

We also found an interaction effect of the experimental conditions on reaction time ($p=0.032$).

Tests of Within-Subjects Contrasts

Measure: MEASURE_1

Source	uncertainty	complexity	Type III Sum of Squares	df	Mean Square	F	Sig.
uncertainty	Linear		.390	1	.390	12.420	.002
Error(uncertainty)	Linear		.723	23	.031		
complexity		Linear	.049	1	.049	1.663	.210
Error(complexity)		Linear	.683	23	.030		
uncertainty * complexity	Linear	Linear	.123	1	.123	5.219	.032
Error (uncertainty*complexity)	Linear	Linear	.541	23	.024		

This result has been included in the appendix (Section 1).

The authors now include some basic, descriptive data about the subject's performance in the task. Figure S2B shows data for L/R choice in stage 1 as well as a measure of choice consistency in stage 2. However, the behavioral data shown in Figure S2B are collapsed across all conditions of complexity and uncertainty. The authors argue this is valid for the stage 1 decision because complexity is not introduced until stage 2. But rather than speculate or make assumptions, behavior could be quantified. This begins to give us an idea of the pattern of choice indicative of MB control, and therefore choice optimality, but this is only a start.

>> [Reply 1-2]

Although we have fully addressed the choice behavior issue in our reply 1-1, for the sake of full clarification, following the reviewer's suggestion, we now report choice behaviors plotted for each trial in the decision tree across all conditions of complexity and uncertainty.

However, we do not think that showing disaggregated behavioral data at various stages of the task are especially informative. We are of course happy to include a file containing these plots as an appendix to the manuscript (indeed we have already included the full trial-by-trial raw behavioral data for consideration by reviewers and this data will be made available to all readers when the manuscript is published). However, simply plotting these effects are in our opinion not informative because they are not diagnostic of the difference between model-based and model-free control, which is the whole point of the manuscript. Given that we have a theoretically justified rationale for how model-based and model-free agents should behave, the behavioral analyses that we report should be motivated by our theoretical predictions. Simply plotting raw behavior as a function of various stages of task performance doesn't by itself show anything unless one is specifically looking for the unique signatures of model-based and model-free behavior. For this one needs to look at certain trials and specific instances where the two algorithms are predicted to generate different behaviors. This is what we are doing in the analyses we report above -- carefully curating and plotting comparisons between trials for which there are theoretically predicted differences in the behavior of the controllers. The plots requested do not follow these principles, and are thus in our opinion confusing rather than useful for readers. Nevertheless, we are happy to include the appendix as an additional file to be published alongside the manuscript at the editor and reviewer's discretion.

Below we reproduce all the requested plots below for the reviewer's information. We also include the Appendix in which each of these plots are reproduced.

Choice behavior in stage 1

- choice bias: There is a left choice bias because in the left branch, all the main choices (L1,R1) are associated with coins, whereas in the right branch, only one main choice (R1) is associated with a coin. This indicates that subjects made choices based on value learning.

- choice consistency: We found a main effect of complexity on choice consistency.

Tests of Within-Subjects Contrasts

Measure: MEASURE_1

Source	UNCERTAINTY	COMPLEXITY	Type III Sum of Squares	df	Mean Square	F	Sig.
UNCERTAINTY	Linear		.004	1	.004	1.078	.310
Error(UNCERTAINTY)	Linear		.089	23	.004		
COMPLEXITY		Linear	.044	1	.044	12.935	.002
Error(COMPLEXITY)		Linear	.078	23	.003		
UNCERTAINTY * COMPLEXITY	Linear	Linear	.003	1	.003	.492	.490
Error(UNCERTAINTY*COMPLEXITY)	Linear	Linear	.156	23	.007		

Choice behaviors in stage 2

- Choice bias (left: state2, right, state3) : All the choice preference reflects value bias. Note that the choice bias is not a straightforward measure for quantifying the relationship between experimental conditions and behavior.

- choice consistency: We found main effects of uncertainty and complexity.

Tests of Within-Subjects Contrasts

Measure: MEASURE_1

Source	UNCERTAINTY	COMPLEXITY	Type III Sum of Squares	df	Mean Square	F	Sig.
UNCERTAINTY	Linear		.030	1	.030	5.633	.026
Error(UNCERTAINTY)	Linear		.122	23	.005		
COMPLEXITY		Linear	1.263	1	1.263	97.093	.000
Error(COMPLEXITY)		Linear	.299	23	.013		
UNCERTAINTY * COMPLEXITY	Linear	Linear	1.284E-7	1	1.284E-7	.000	.996
Error(UNCERTAINTY*COMPLEXITY)	Linear	Linear	.092	23	.004		

These results have been included in the appendix (Section 2).

A full analysis must include descriptions of subject choice (number, percentage, and latency) at each stage for each of the four conditions. For example, when choice optimality is high vs low, what does choice behavior look like in the four separate conditions? Is it identical?

>> [Reply 1-3]

To fully accommodate what the reviewer requested, here we provide a full behavioral analysis including the number/percentage and the latency of choice for each of the four conditions and for the level of choice optimality (low vs high). Note that behavioral analyses as a function of choice optimality is valid for the second stage only due to the task design in which all choices in the first stage are optimal in a high uncertainty condition.

1. Percentage of each choice

Subjects exhibit different choice patterns in each of the four separate conditions and for high vs. low choice optimality. Fully consistent with the results of the choice optimality analysis, the number of non-optimal choices increases in the high uncertainty conditions.

- State 2

- State 3

2. Latency (reaction-time): We found that an interaction effect of experimental conditions on RT is stronger when choice optimality is high than when it's low.

- Shown below are RT across all the experimental conditions for the low vs. high choice optimality.

Tests of Within-Subjects Contrasts **optimal**

Measure: MEASURE_1

Source	uncertainty	complexity	Type III Sum of Squares	df	Mean Square	F	Sig.
uncertainty	Linear		.441	1	.441	14.081	.001
Error(uncertainty)	Linear		.721	23	.031		
complexity		Linear	.043	1	.043	1.447	.241
Error(complexity)		Linear	.690	23	.030		
uncertainty * complexity	Linear	Linear	.166	1	.166	5.231	.032
Error (uncertainty*complexity)	Linear	Linear	.731	23	.032		

Tests of Within-Subjects Contrasts **not optimal**

Measure: MEASURE_1

Source	uncertainty	complexity	Type III Sum of Squares	df	Mean Square	F	Sig.
uncertainty	Linear		.325	1	.325	5.400	.029
Error(uncertainty)	Linear		1.383	23	.060		
complexity		Linear	.120	1	.120	1.611	.217
Error(complexity)		Linear	1.716	23	.075		
uncertainty * complexity	Linear	Linear	.109	1	.109	4.259	.051
Error (uncertainty*complexity)	Linear	Linear	.588	23	.026		

In summary, the choice behavior (number/percentage and latency) reflects choice optimality. These results have been included in the appendix (Section 3).

Independent of choice optimality, how do uncertainty and complexity affect reaction time and choice behavior in each of the four conditions?

>> [Reply 1-4]

1. Percentage of each choice: Subjects prefer choices that are not associated with zero-value outcome state (state 4 and 10).

- State 1

- State 2

- State 3

2. Latency (reaction-time): We found an interaction effect of the experimental conditions on RT for the second stage.

Tests of Within-Subjects Contrasts

Measure: MEASURE_1

Source	uncertainty	complexity	Type III Sum of Squares	df	Mean Square	F	Sig.
uncertainty	Linear		.005	1	.005	.484	.494
Error(uncertainty)	Linear		.229	23	.010		
complexity		Linear	.016	1	.016	2.633	.118
Error(complexity)		Linear	.144	23	.006		
uncertainty * complexity	Linear	Linear	.044	1	.044	1.683	.207
Error (uncertainty*complexity)	Linear	Linear	.608	23	.026		

Tests of Within-Subjects Contrasts

Measure: MEASURE_1

Source	uncertainty	complexity	Type III Sum of Squares	df	Mean Square	F	Sig.
uncertainty	Linear		.390	1	.390	12.420	.002
Error(uncertainty)	Linear		.723	23	.031		
complexity		Linear	.049	1	.049	1.663	.210
Error(complexity)		Linear	.683	23	.030		
uncertainty * complexity	Linear	Linear	.123	1	.123	5.219	.032
Error (uncertainty*complexity)	Linear	Linear	.541	23	.024		

In summary, despite the fact that these are very simple and noisy measures, we were still able to find effects of experimental conditions, albeit in some cases these effects do not reach statistical significance, likely because we are splitting the data in ways that were never intended given our experimental design, thereby substantially reducing our statistical power to detect meaningful effects.

These results have been included in the appendix (Section 4).

Further, what is the relationship between the observed neural signatures of arbitration and the subject's actual choice behavior? This is not trivial. Throughout the discussion, the authors link neural processes of arbitration to behavior. These are just excerpts from first paragraph: These behavioral findings were supported by evidence that a region of the brain previously implicated in the arbitration process...", "Taken together, these findings help to advance our understanding of the contribution of two key variables to the arbitration process at behavioral and neural levels."

Yet, the authors have only actually linked neural signatures to choice optimality, which asks if the choice behavior pattern matches that of the ideal MB agent. MB and MF strategies are ultimately just constructs that allow us to explain patterns of choice behavior. Linking neural signatures of arbitration to actual choice behavior is key to the goals of this study. As it stands, this link has not conclusively been made.

>> [Reply 1-5]

We respectfully disagree with the reviewer on his/her assessment. We have a theoretical model that specifies how we theorize the arbitration process between model-based and model-free RL would function. This model suggests that the reliability of the two strategies which are estimated based on the prediction errors incurred in the two systems should drive the arbitration process, in combination with a consideration about the complexity (which is the new addition to the model beyond our original 2014 paper). We have linked our theoretical model of arbitration (which best explains behavior on the task), with the brain activity. Thus we have directly linked the model to both behavior and neural substrates. We are not sure what else the reviewer is envisaging with regard to his/her request to link neural signatures to actual choice behavior. This linking is done via the theoretical model that we have specified, and the formal model comparison conducted BOTH on the behavioral data and the neural data formalizes and provides direct evidence for this link.

Given that we have used choice optimality as a non-model-based index of model-based control, we have also now conducted an additional analysis in which we show evidence for choice optimality in the iIPFC ROI that we also identified as showing the computational signatures of the arbitration process. Consistent with the relationship between our computational model predictions, computational regressors found to be correlated with activity in the brain, and the relationship between choice optimality and model-based predictions, here we show that choice optimality is also reflected in our ROIs. Shown is a comparison between the trials in which choice optimality is high and low (the exact p-values for the contrast are shown above each plot).

This result has been included in the supplementary information (Figure S8).

Figure S8. Given that we have used choice optimality as a non-model-based index of model-based control (Figure 3), we have also now conducted an additional analysis in which we show evidence for choice optimality in the iIPFC ROI that we also identified as showing the computational signatures of the arbitration process. Consistent with the relationship between our computational model predictions, computational regressors found to be correlated with activity in the brain, and the relationship between choice optimality and model-based predictions, here we show that choice optimality is also reflected in our ROIs. Shown is a comparison between the trials in which choice optimality is high and low. This is consistent with the notion that when model-based control is increased, there is an increased activation in the vIPFC associated with an increased engagement of model-based control, and a decreased engagement of model-free control (the exact p-values for the contrast are shown above each plot). Error bars are SEM across subjects.

Reviewer #2 (Remarks to the Author):

Overall, the authors performed a thorough revision and were responsive to the critique raised. I do have a few rather minor comments.

Re Reply 2-(1), (i) Remarks on choice repetition: At the end of their answer the authors conclude 'Thus in our study, the application of the choice consistency measure is confined to a simple readout of reward-based value learning guided by model-free control'. This is actually what I asked for ('which could constitute a measure of model-free control'). I do see why measures of choice repetition / consistency are not well applicable for model-based control and also do not to indicate the arbitration between MF and MB. Thus, I do appreciate the authors' presentation of choice optimality. The reports of choice optimality are overall convincing for indicating the superiority of model-based control in this particular environment and the effect of the manipulations that do affect on arbitration. However, it does not provide a distinct behavioural readout because it assumes that one strategy is simply more optimal than the other which substantially relies upon how the environment is designed. You could also design a task the way that MF would be more optimal. However, for some measure of model-free control, I think it is possible to demonstrate it as outlined before (and the authors actually do that in the supplement). Also by using simulations of a model-free learner only, it should be possible to demonstrate this even clearer (once again this seems to be done in the supplement now). I think the supplemental analysis presented in S-Figure 2 is quite interesting and should be moved to the main manuscript (choice bias and choice consistency).

>> [Reply 2-1]

We greatly appreciate the reviewer's effort to lend valuable insight into our study. Yes, we agree that choice optimality is a good behavioral indicator of model-based control, albeit in the limited sense.

As the reviewer pointed out, it would be very interesting to design a task that allows us to discover clearer readouts for model-based and model-free control, combined with the simulations of a pure model-free learner. We think that properly addressing this issue is critical and requires another set of experiments/analyses/simulations, so it would be better to leave it for a future study, rather than providing preliminary results as a supplement.

To fully incorporate the reviewer's suggestion, we have moved Figure S2 to the main manuscript. Figure S2A and 2B, behavior of subjects, have been moved to Figure 2A and 2B; Figure S2C, prediction of the computational model, has been moved to Figure 6A.

I find it an important demonstration of the behaviour going on in this task. To be able to incorporate this, the choice optimality section could be shortened a bit or parts be moved to the supplement (it is a bit lengthy at some point).

>> [Reply 2-2] Thanks a lot for the good suggestion. We have shortened the choice optimality section in the revised version of our manuscript, moving details to the supplement.

Main text:

"Behavioral measure (choice optimality). Choice optimality measure quantifies the extent to which participants on a given trial took the objectively best choice had they complete access to the task state-space, and a perfect ability to plan actions in that state-space. It is based on the choice of the ideal agent assumed to have a full, immediate access to information of the environmental structure, including state-transition uncertainty and task complexity. The choice optimality is defined as the degree of match between subjects' actual choices and an ideal agent's choice corrected for the number of available options. To compute the degree of choice match between the subject and the ideal agent,

for each condition, we calculated an average of normalized values (i.e., likelihood) of the ideal agent for the choice that a subject actually made on each trial. To correct for the number of options, we then multiplied it by 2 for the high complexity condition; this is intended to compensate for the effect that the baseline level of the likelihood in the high complexity condition (# of available options =4) becomes half of that in the low complexity condition (# of available options =2). In other words, this adjustment effectively compensates the effect of # of available options on normalization without biasing the correspondence between participant's choices and optimal choices. The choice optimality value would have a maximum/minimum value if a subject made the same/opposite choice as the ideal agent's in all trials, regardless of complexity condition changes.

Full details of choice optimality are provided in Supplementary Methods - Behavioral measure (choice optimality)."

Behavior and parameter recovery make an important contribution. I value them much more than the large scale model comparison.

>> **[Reply 2-3]** Again we thank the reviewer for providing constructive suggestions, which have greatly improved the manuscript.

For the fMRI analysis, I think some more specification of the GLM needs to go to the main manuscript.

>> **[Reply 2-4]** We have moved the GLM section, including its detailed specification, to the main manuscript.

"Finally, we implemented a second-level random effects analysis for each regressor of interest..." what does that mean exactly? Where they all included in one second-level random effects model or each in a separate? Please specify and explain the reasoning.

>> **[Reply 2-5]** We apologize for the confusion. We mean it by a standard second level GLM analysis. We first ran the 1st level GLM including all the regressors of interest, and then ran a one-sample t-test at the 2nd level for each separate regressor (i.e., random effects model each in a separate). We have clarified this in the main manuscript (Experimental Procedures- GLM design section; see below).

"Finally, we implemented a standard second-level random effects analysis for each regressor of interest, and applied correction for multiple comparisons. Specifically, after running the 1st level GLM including all the regressors of interest, we ran a one-sample t-test at the 2nd level for each separate regressor (i.e., random effects model each in a separate)."

Again, we greatly appreciate the reviewer's very insightful and constructive comments, which have made it possible for us to significantly improve our paper.

REVIEWERS' COMMENTS:

Reviewer #1 (Remarks to the Author):

The concerns have been adequately addressed. Thank you.

Reviewer #2 (Remarks to the Author):

The authors have thoroughly responded to my questions. It's an interesting paper - congratulations! There will be ongoing debate what is the best way to validate behavioral measures and it remains a challenge to develop tasks with specific and valid model-free vs. model-based readouts.